# REWARD MODEL ROUTING IN ALIGNMENT

**Xinle Wu, Yao Lu**
National University of Singapore
{wuxl,luyao}@comp.nus.edu.sg

## ABSTRACT

Reinforcement learning from human or AI feedback (RLHF/RLAIF) has become the standard paradigm for aligning large language models (LLMs). However, most pipelines rely on a single reward model (RM), limiting alignment quality and risking overfitting. Recent work explores RM routing—dynamically selecting an RM from a candidate pool to exploit complementary strengths while maintaining $O(1)$ RM calls—but existing methods suffer from cold-start and insufficient exploration. We propose BayesianRouter, a hybrid routing framework that combines offline RM strengths learning with online Bayesian selection. In the offline stage, a multi-task router is trained on preference data to estimate per-RM reliability. In the online stage, a Bayesian Thompson sampling router performs per-query RM selection, initializing RM-specific weight vectors with offline embeddings as Gaussian priors and adaptively updating their posteriors with online rewards to adapt to the evolving policy distribution. Extensive experiments on instruction-following (AlpacaEval-2, Arena-Hard, MT-Bench) and reasoning (GSM8K, MMLU) benchmarks show that BayesianRouter consistently outperforms individual RMs, RM ensembling, and existing routing methods. [1]

## 1 INTRODUCTION

Large language models (LLMs) have revolutionized artificial intelligence, demonstrating substantial capabilities in language understanding, reasoning, and open-ended text generation across diverse domains (Guo et al., 2025; Achiam et al., 2023). To safely and effectively deploy LLMs, recent post-training techniques, particularly reinforcement learning from human feedback (RLHF) and its AI-augmented variant (RLAIF) (Wang et al., 2024b), aimed at aligning LLMs with human values and preferences. These methods fine-tune LLMs to internalize nuanced human preferences, bridging the gap between raw pretrained performance and user-aligned behavior. In standard RLHF, a reward model (RM) provides the feedback signal for optimizing the policy LLM; for example, in PPO, the RM provides a scalar reward to directly increase the probability of preferred responses (Ziegler et al., 2019), while in direct preference optimization (DPO), the RM compares two candidate responses to determine which is better (Dong et al., 2024).

Recent RLHF/RLAIF pipelines often rely on a single RM throughout training (Kaufmann et al., 2024). This design choice, however, can be suboptimal due to (1) limited generalizability: no single RM consistently excels across all tasks, as evidenced by benchmarks like RewardBench 2 (Malik et al., 2025). An RM tuned for one type of content (e.g. conversational helpfulness) may perform poorly on a different genre (e.g. mathematical reasoning), leading to suboptimal alignment when one fixed RM is used universally; (2) high costs: using a powerful general-purpose LLM (e.g. GPT-5) as the RM can provide high-quality feedback, but the cost of querying such a model at scale is prohibitive (Zheng et al., 2023). In practice, this makes large RMs impractical for extensive RLHF training, and (3) risk of overoptimization: relying on a single RM amplifies the risk of overfitting to that RM's idiosyncratic biases or noise, which can lead the policy to exploit the RM's flaws (i.e., reward hacking) rather than truly align with human intent (Coste et al., 2023; Zhang et al., 2024). Collectively, these issues undermine the robustness and scalability of single-RM alignment.

To overcome these challenges, recent studies have started to leverage an ensemble of reward models, combining the strengths of multiple RMs. Prior work in (Coste et al., 2023; Zhang et al., 2024)

---

[1]Code is available at https://github.com/XinleWu/BayesianRouter.

explored multi-RM approaches, but naively using multiple RMs in parallel for every query is extremely costly and can introduce conflicting or noisy signals when the models disagree. Among ensemble strategies, routing methods are particularly interesting: instead of aggregating all models' outputs, a router dynamically selects the most suitable RM for each input, preserving the benefits of model diversity while minimizing overhead. In this spirit, LASER (Nguyen et al., 2024) is, to our knowledge, the first method to apply instance-level RM routing in RLHF. LASER frames reward model selection as a contextual multi-armed bandit problem. During DPO-based RLHF training, for each batch of prompts, a bandit (LinUCB) chooses a single RM from a candidate pool to label the policy's responses; the policy is then updated on this preference-labeled data, and the router is updated based on the policy's resulting reward signal. By selecting one RM at a time, LASER avoids the overhead of running all RMs and adapts the choice of RM as training progresses.

While LASER showed the promise of adaptive RM selection, important limitations remain: (1) coarse-grained routing: LASER selects one RM per batch of prompts, whereas prompts within the same batch may favor different RMs. This batch-level routing often makes suboptimal choices for many individual queries; (2) limited exploration: using LinUCB (which relies on point estimates and optimism), the router can prematurely lock onto a suboptimal RM and insufficiently explore others. In other words, LASER's bandit may over-exploit one arm without adequately probing alternatives that could be better for certain query types, and (3) inefficient cold-start: at the start of training, LASER assumes all RMs are equally good and must gather many interactions to identify each RM's unique strengths. This slow start results in suboptimal RMs being used in early training, which reduces sample efficiency and makes the outcome sensitive to initial conditions.

To address these issues, we propose BayesianRouter, a hybrid RM routing framework that integrates a learned model of RM strengths, paired with an online Bayesian selection strategy to accelerate routing at a small compute overhead. Specifically, BayesianRouter consists of (1) an offline router with a language model-based encoder that is trained on existing preference datasets to predict which RMs will perform better for a given query. We use a multi-task objective: a Bradley–Terry ranking head scores each candidate RM, and a classification head predicts whether each RM would choose the better answer in a given pair. This offline router captures each RM's specialization in a shared embedding space, providing a rich prior for selection; (2) online Bayesian router: during RLHF fine-tuning, BayesianRouter employs a Bayesian Thompson sampling for instance-level RM selection. The router treats the query embedding as context and maintains a Gaussian posterior for each RM's reward model. For each query, it samples a reward estimate for each RM from its posterior and selects the RM with the highest sample, then uses that RM's feedback to train the policy and update the posterior. By sampling from an uncertainty-aware model, instead of relying on a single deterministic estimate, the router naturally balances exploration and exploitation and can more robustly discover which RM is optimal for each query type. We initialize the online router using the prior knowledge from the offline router to inherit the knowledge on each RM's strengths, as well as to bootstrap the cold-start. As training proceeds, the router updates this knowledge and adapts to the evolving policy distribution while retaining the offline insights.

We evaluate BayesianRouter on both instruction-following benchmarks (including AlpacaEval-2 and MT-Bench) and academic benchmarks (including GSM8K and MMLU). The results demonstrate that BayesianRouter significantly outperforms strong baselines, such as the single best RM, RM ensemble methods, and LASER.

## 2 RELATED WORK

**LM-based Reward Model.** Language model-based reward models (RMs) act as proxies for human preferences and play a central role in RLHF and RLAIF (Kaufmann et al., 2024; Wang et al., 2024b). They are commonly categorized into three families: classifier RMs (Liu et al., 2025), generative RMs (Yu et al., 2025a), and LLM-as-a-judge (LAJ) (Hurst et al., 2024). In early RLHF pipelines, RMs typically provided a scalar reward for each (prompt, response), which was then optimized with policy-gradient methods such as PPO (Kaufmann et al., 2024). More recent RLAIF approaches often use RMs to conduct pairwise comparisons between responses and apply objectives such as DPO to encourage the policy to prefer the better response (Guo et al., 2024). Research on RMs primarily focuses on improving the reliability of reward signals. For example, (Wang et al., 2024a) filter unreliable preference data by comparing rankings across different training iterations;

(Yu et al., 2024) adopt a divide-and-conquer strategy that decomposes response evaluation into simpler claim-level judgments; and (Liu et al., 2025) build a large-scale dataset of 40M preference pairs via human–AI collaboration, enabling smaller RMs to outperform much larger models. Orthogonally, RM ensembling (e.g., averaging, lower-confidence bound, or uncertainty-weighted schemes) has been shown to improve robustness and mitigate overoptimization risks (Coste et al., 2023; Zhang et al., 2024). In addition, benchmarks such as RewardBench, RM-Bench, and RewardBench 2 provide systematic evaluations of different RMs across domains, offering practical guidance for model selection (Lambert et al., 2024; Malik et al., 2025; Liu et al., 2024b).

**Routing LLM Queries.** Research on routing LLM queries has so far mainly focused on *LLM inference*, where the aim is to assign each query to the most suitable model before decoding in order to balance accuracy and efficiency. For instance, Lu et al. (2023) train a router using reward-model-based scores of candidate responses as supervision; Ding et al. (2024) adaptively switch between cloud and edge models depending on query difficulty; Ong et al. (2024) propose ROUTELLM, which employs classifiers (e.g., matrix factorization, causal LLM classifier) to decide whether a query should be routed to a strong or weak model; Shadid et al. (2025) analyzes LLM performance on benchmark tasks, clusters user queries by similarity, and dynamically routes each query to the best-performing LLM for its cluster, achieving higher accuracy at lower cost compared to trained routers; and Frick et al. (2025) introduce P2L, which uses Bradley–Terry modeling and sparse pairwise preference data to train routers that scale to hundreds of candidate LLMs. While these methods are designed for inference scenarios, to the best of our knowledge, there has not been work on leveraging *offline preference data* to pretrain a router specifically for reward models, which differ from inference routers in terms of input features and label construction. Moreover, a purely offline router is prone to out-of-distribution (OOD) issues, and may generalize poorly when deployed on unseen data distributions.

**Multi-Armed Bandits (MABs).** MABs offer a classical framework for sequential decision-making under uncertainty, balancing exploration and exploitation by pulling one arm per round and observing stochastic feedback (Zhou, 2015; Bouneffouf & Rish, 2019). This formulation naturally fits routing, where each query is assigned to one model with only partial feedback available. Among contextual bandit algorithms, LinUCB (Li et al., 2010) and Bayesian linear Thompson sampling (Agrawal & Goyal, 2013) are two widely used approaches: the former uses optimism via confidence bounds, while the latter samples from a posterior to enable uncertainty-aware exploration. Beyond these, variants like KL-UCB (Garivier & Cappé, 2011), OFUL (Abbasi-Yadkori et al., 2011), or logistic/GLM bandits (Filippi et al., 2010) can also be applied depending on feedback type and distributional assumptions.

MAB algorithms have recently been applied to *LLM inference routing*. For example, (Li, 2025) formulates model selection as a contextual bandit problem, training preference-conditioned dynamic routing policies on offline data and leveraging model identity embeddings to generalize across architectures, thereby enabling adaptive selection of high-performance, low-cost LLMs at inference time. In the *reward modeling* setting, the only existing MAB-based router is LASER (Nguyen et al., 2024), which leverages LinUCB to select one reward model per input batch during RLAIF training. In contrast, our BayesianRouter replaces point-estimate exploration with Bayesian posterior sampling, and further integrates offline-learned priors to address both exploration inefficiency and cold-start limitations.

## 3 METHODS

In this section, we introduce BayesianRouter, a reward model (RM) routing framework designed for adaptive RM selection within preference-based alignment pipelines. BayesianRouter is designed for the DPO family, while it could in principle also benefit reinforcement learning–based methods such as PPO that rely on scalar reward signals, although evaluating its performance in that setting is beyond the scope of this work. Concretely, for each input *preference pair*—consisting of a prompt and two candidate responses from the policy model—BayesianRouter selects the most appropriate RM from a candidate pool to evaluate the pair. The resulting preference signal is then used to train the policy model via online DPO. An overview of BayesianRouter is shown in Figure 1.

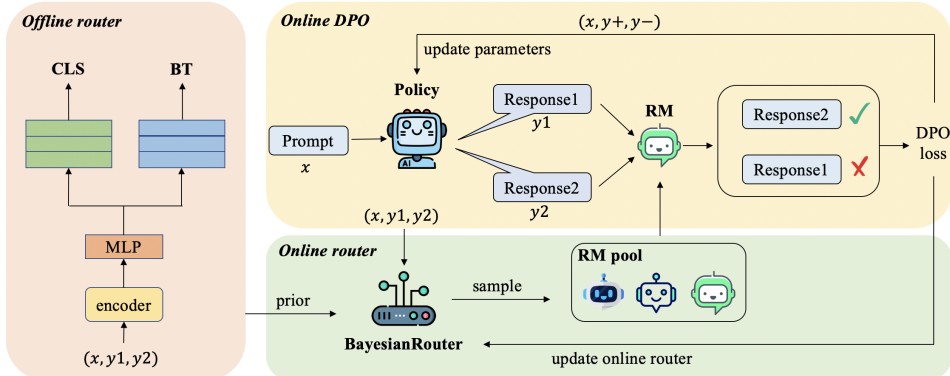

Figure 1: Overview of BayesianRouter.

We structure this section as follows. Section 3.1 briefly reviews the standard online DPO algorithm, formally defines the RM selection problem, and provides an overview of BayesianRouter. Section 3.2 introduces the offline RM router, which leverages preference datasets and a multi-task objective to model RM strengths, i.e., identifying which candidate RM is most reliable for a given preference pair. Section 3.3 presents the Bayesian Thompson sampling–based online router, which adaptively selects RMs during online DPO training and updates its posterior distribution with policy feedback. Finally, Section 3.4 describes how we integrate the offline-learned RM strengths with the online router to alleviate the cold-start problem.

## 3.1 PROBLEM FORMULATION AND METHOD OVERVIEW

**Online DPO Training Pipeline.** We follow the standard online Direct Preference Optimization (DPO) (Guo et al., 2024; Dong et al., 2024) setup to iteratively align the policy model $\pi$. Let $\pi_t$ denote the model at training step $t$, initialized from a pretrained policy $\pi_0$. At each step, the model receives a mini-batch of prompts $\{x_i\}_{i=1}^B$, and for each $x_i$ we sample $k$ candidate responses $y_i = \{y_i^1, \dots, y_i^k\} \sim \pi_m(\cdot|x_i)$. A reward model $R$ evaluates these candidates to construct preference pairs: for classifier RMs, $R(x_i, y_i^j)$ outputs a scalar score and we form a pair $(x_i, y_i^w, y_i^l)$ if $R(x_i, y_i^w) > R(x_i, y_i^l)$; for generative RMs, the model directly compares two responses $(y_i^j, y_i^{j'})$ and indicates which is preferred, yielding a preference pair of the same form. Collecting over the batch gives a preference dataset $\mathcal{D}_{\text{pref}} = \{(x_i, y_i^w, y_i^l)\}_{i=1}^B$.

The policy $\pi_t$ is then updated on $\mathcal{D}_{\text{pref}}$ using the DPO loss (Rafailov et al., 2023), which encourages the policy to increase the likelihood ratio between the preferred and dispreferred responses relative to a frozen reference model $\pi_{\text{ref}}$:

$$\mathcal{L}_{\text{DPO}} = -\frac{1}{|\mathcal{D}_{\text{pref}}|} \sum_{(x, y^w, y^l) \in \mathcal{D}_{\text{pref}}} \log \sigma \left( \beta \log \frac{\pi_t(y^w \mid x)}{\pi_{\text{ref}}(y^w \mid x)} - \beta \log \frac{\pi_t(y^l \mid x)}{\pi_{\text{ref}}(y^l \mid x)} \right), \quad (1)$$

where $\pi_{\text{ref}}$ is typically set to $\pi_0$, $\sigma(\cdot)$ is the logistic function and $\beta$ is a scalar hyperparameter. Without loss of generality, we use the standard DPO objective, though other variants such as IPO (Azar et al., 2024) and SLiC (Zhao et al., 2023) are also compatible with our framework.

**Problem Formulation.** In the online DPO pipeline, the choice of the reward model (RM) directly shapes the mini-batch preference dataset $\mathcal{D}_{\text{pref}}$ constructed at each step, thereby determining the quality of the alignment signal. Empirically, no single RM uniformly dominates across preference types or domains. For example, RewardBench 2 (Malik et al., 2025) reports that `Skywork-Reward-V2-Llama-3.1-8B` ranks first overall but, while outperforming the second-place `LMUnit-qwen2.5-72b` on Math and Safety preferences, it underperforms `LMUnit-qwen2.5-72b` on Factuality. To exploit complementary strengths across $N$ RMs, one option is to ensemble them (e.g., majority voting over multiple RMs). However, this increases the per-step inference cost from $O(1)$ to $O(N)$ RM calls, resulting in a significant increase in training

cost. In contrast, we adopt *per-query routing*: for each unlabeled preference pair $(x_i, y_i^j, y_i^{j'})$, we select a *single* most suitable RM to annotate it. This preserves $O(1)$ RM calls per step while still leveraging complementary strengths of $N$ RMs.

Formally, let $\mathcal{M} = \{R_n\}_{n=1}^N$ be the candidate RM pool. We aim to learn a router $\mathrm{Router}(\cdot\,; W)$ (parameterized by $W$) that maps an unlabeled preference pair $(x_i, y_i^j, y_i^{j'})$ to an RM:

$$R_n = \mathrm{Router}\big(x_i,\, y_i^j,\, y_i^{j'}\,; W\big).$$

The selected RM yields more reliable preference annotations for the current batch, thereby providing higher-quality supervision for policy updates.

Routers can be *offline* (trained on a static preference corpus and kept fixed during online DPO) or *online* (updated during training using feedback from the current policy). Offline routers can leverage existing labeled offline preference datasets but may fail under distribution shift between the offline training data and the online data; online routers can adapt to the target distribution but suffer from cold-start and exploration challenges early in training.

**Our Approach: BayesianRouter (Overview).**    We propose BayesianRouter, a hybrid RM routing framework that couples an *offline, RM strengths learning* stage with an *online, distribution adaptation* stage (see Fig. 1). The offline-learned RM strengths provide a high-quality initialization that mitigates the cold-start issue and improves early batch-level routing accuracy; the Bayesian online updates adapt the router to the evolving data distribution and policy. The following subsections detail the offline router, the online RM router, and the integration strategy.

## 3.2    OFFLINE RM ROUTER

Given a candidate set of reward models $\mathcal{M} = \{R_n\}_{n=1}^N$ and an offline preference dataset $\hat{\mathcal{D}}_{\mathrm{pref}} = \{(x_i, y_i, y_i', \ell_i)\}_{i=1}^m$ where $\ell_i \in \{0, 1\}$ indicates which of $y_i$ and $y_i'$ is the preferred response to the prompt $x_i$, the goal of the *offline router* is to predict, for a new unlabeled preference pair $(x, y, y')$, which RM is most likely to correctly identify the preferred response. Figure 1 (left) illustrates the architecture of our offline RM router.

**Collecting RM Behavior Data.**    To construct the training signals for the offline router, we first collect the behavior of each candidate RM on $\hat{\mathcal{D}}_{\mathrm{pref}}$. Concretely, we run each RM $R_n$ on each preference pair $q_i = (x_i, y_i, y_i') \in \hat{\mathcal{D}}_{\mathrm{pref}}$ and record a binary indicator $\delta_i^{(n)} \in \{0, 1\}$ that equals 1 if $R_n$ agrees with the ground truth $\ell_i$ and 0 otherwise, yielding $\mathcal{D}_{\mathrm{beh}} = \{(q_i, \delta_i^{(n)}) \mid i = 1, \ldots, m;\ n = 1, \ldots, N\}$.

**Preference-pair feature construction.**    Unlike LASER in (Nguyen et al., 2024) that uses only the prompt as router input, we encode the whole preference pair $(x_i, y_i, y_i')$ because an RM's decision depends not only on the prompt but also on the semantic content of the two responses and their contrast. Concretely, we first concatenate the prompt with each response and encode them with a shared pretrained encoder $\mathrm{Enc}(\cdot\,; W_e)$:

$$\mathbf{e}_i = \mathrm{Enc}(x_i \,\|\, y_i; W_e), \qquad \mathbf{e}_i' = \mathrm{Enc}(x_i \,\|\, y_i'; W_e).$$

We then aggregate these encodings into a single preference-pair representation by

$$\mathbf{h}_i = \mathrm{MLP}\big(\,[\mathbf{e}_i + \mathbf{e}_i';\ |\mathbf{e}_i - \mathbf{e}_i'|]\,; W_l\big) \in \mathbb{R}^d,$$

where $[\cdot\,; \cdot]$ denotes vector concatenation, $|\cdot|$ is the element-wise absolute difference, and $\mathrm{MLP}(\cdot\,; W_l)$ is a single-layer MLP used to fuse features.

Based on the preference feature $\mathbf{h}_i$, we adopt a multi-task objective with two prediction heads. The primary head is a Bradley–Terry (BT) head, which assigns an *ability score* to each RM such that, given a preference pair, the RM with the highest score is selected as the most reliable one. The auxiliary head is a classification (CLS) head, which independently predicts for each RM whether it can correctly identify the preferred response in the given pair.

**BT head (primary).**    We define a *disagreement sample* as a preference pair on which two RMs produce different behavior labels. Such samples capture the relative competence of the two RMs

and are therefore suitable for training a Bradley–Terry (BT) head to predict per-RM ability scores. Formally, from $\mathcal{D}_{\text{beh}}$ we extract the disagreement set $\mathcal{D}_{\text{bt}} = \{(q_i, n, n') \mid \delta_i^{(n)} = 1, \delta_i^{(n')} = 0\}$. We learn an embedding matrix $E_{\text{bt}} \in \mathbb{R}^{N \times d}$ whose $n$-th row represents RM $R_n$. We compute BT scores as inner products, i.e., $s_i^n = \langle \mathbf{h}_i, E_{\text{bt}}[n] \rangle$ and $s_i^{n'} = \langle \mathbf{h}_i, E_{\text{bt}}[n'] \rangle$, and optimize the pairwise logistic (Bradley–Terry) loss

$$\mathcal{L}_{\text{bt}} = -\frac{1}{|\mathcal{D}_{\text{bt}}|} \sum_{(q_i, n, n') \in \mathcal{D}_{\text{bt}}} \log \sigma\big(s_i^n - s_i^{n'}\big). \tag{2}$$

This objective encourages the BT head to assign higher ability scores to RMs that win paired comparisons.

**CLS head (auxiliary).** Since the BT head relies only on disagreement samples, it ignores the information contained in the remaining portion of $\mathcal{D}_{\text{beh}}$. To better exploit the full dataset, we introduce an auxiliary per-RM binary classification head. Given a preference pair $q_i$, the CLS head predicts each candidate RM's behavior label $\delta_i^{(n)}$ from the preference embedding $\mathbf{h}_i$. Concretely, we learn an embedding matrix $E_{\text{cls}} \in \mathbb{R}^{N \times d}$ whose $n$-th row corresponds to RM $R_n$, compute logits $z_i^n = \langle \mathbf{h}_i, E_{\text{cls}}[n] \rangle$ for every RM, and optimize the binary cross-entropy over $\mathcal{D}_{\text{beh}}$:

$$\mathcal{L}_{\text{cls}} = -\frac{1}{|\mathcal{D}_{\text{beh}}|} \sum_{(q_i, n) \in \mathcal{D}_{\text{beh}}} \Big[\delta_i^{(n)} \log \sigma(z_i^n) + (1 - \delta_i^{(n)}) \log \big(1 - \sigma(z_i^n)\big)\Big]. \tag{3}$$

By independently predicting each RM's behavior, the CLS head provides complementary supervision to the pairwise BT objective, benefiting the BT ranking through shared representation learning.

**Training objective and offline output.** The router is trained by minimizing the combined loss $\mathcal{L}_{\text{total}} = \mathcal{L}_{\text{bt}} + \lambda \mathcal{L}_{\text{cls}}$, where $\lambda$ controls the contribution of the auxiliary classification loss. We optimize the encoder parameters $W_e$, the MLP parameters $W_l$, and the head parameters $E_{\text{bt}}$ and $E_{\text{cls}}$. After training, we retain the BT embedding matrix $E_{\text{bt}}$ as the prior for online routing, since it captures relative RM strengths conditioned on preference pairs.

## 3.3 BAYESIAN ONLINE RM ROUTER

Unlike the offline router, which is trained on static preference data and remains fixed during online training, the online router is continuously updated after each routing decision using observed rewards. By adapting to the evolving policy-induced distribution of preference pairs $\mathcal{D}_{\text{pref}}$, the online router mitigates distributional mismatch that would otherwise limit the effectiveness of the offline router.

In online routing, only the supervision from the selected RM is observed for each preference pair, making the problem a natural instance of contextual partial-feedback learning (i.e., a contextual bandit). Here, candidate RMs correspond to arms, the preference pair serves as the context, and the problem can be addressed with contextual multi-armed bandit (MAB) algorithms. (Nguyen et al., 2024) used LinUCB for online routing. However, we empirically find LinUCB often collapses to a single fixed arm after a few batches, likely because per-arm observations are scarce and contexts are similar, which leads to premature exploitation. To encourage continued exploration and to more reliably discover which contexts each RM specializes in, we adopt Bayesian Thompson sampling.

**Bayesian Thompson Sampling** We model the expected utility of selecting RM $R_n$ on a preference-pair embedding $\mathbf{h}_i$ with Bayesian linear regression:

$$r = \mathbf{w}_n^\top \mathbf{h}_i + \varepsilon, \qquad \varepsilon \sim \mathcal{N}(0, \sigma^2), \tag{4}$$

where $\mathbf{w}_n \in \mathbb{R}^d$ is a latent weight vector for $R_n$ and $\sigma^2$ denotes observation noise. Each RM maintains a Gaussian posterior $\mathbf{w}_n \sim \mathcal{N}(\boldsymbol{\mu}_n, \boldsymbol{\Sigma}_n)$ that is updated only when $R_n$ is selected. At training step $t$, given a batch of preference-pair embeddings $\{\mathbf{h}_i\}_{i \in \mathcal{B}_t}$, we perform Thompson sampling by drawing a sample from each RM's posterior for each preference pair:

$$\mathbf{w}_n^{(t)} \sim \mathcal{N}(\boldsymbol{\mu}_n^{(t)}, \boldsymbol{\Sigma}_n^{(t)}), \qquad n_i^* = \arg\max_n \mathbf{h}_i^\top \mathbf{w}_n^{(t)},$$

and the router selects $R_{n_i^*}$ for that pair. Let $\mathcal{I}_n^{(t)} = \{i \in \mathcal{B}_t \mid n_i^* = n\}$ denote the indices in the batch assigned to RM $R_n$; after observing scalar rewards $\{\hat{r}_n^i\}_{i \in \mathcal{I}_n^{(t)}}$ for these pairs, we update $R_n$'s posterior using the accumulated sufficient statistics of its assigned pairs:

$$\boldsymbol{\Sigma}_n^{(t+1)} = \left( \boldsymbol{\Sigma}_n^{(t)\,-1} + \frac{1}{\sigma^2} \sum_{i \in \mathcal{I}_n^{(t)}} \mathbf{h}_i \mathbf{h}_i^\top \right)^{-1}, \tag{5}$$

$$\boldsymbol{\mu}_n^{(t+1)} = \boldsymbol{\Sigma}_n^{(t+1)} \left( \boldsymbol{\Sigma}_n^{(t)\,-1} \boldsymbol{\mu}_n^{(t)} + \frac{1}{\sigma^2} \sum_{i \in \mathcal{I}_n^{(t)}} \hat{r}_n^i \mathbf{h}_i \right). \tag{6}$$

When no offline prior is injected we initialize $\boldsymbol{\mu}_n^{(0)} = \mathbf{0}$ and $\boldsymbol{\Sigma}_n^{(0)} = \sigma_w^2 I_d$, where $\sigma_w^2$ is the prior variance. Let $\mathcal{L}_{\mathrm{DPO}}^i$ be the DPO loss on preference pair $i$ labeled by $R_n$. We follow LASER to take the raw reward to be $\tilde{r}_n^i = -\mathcal{L}_{\mathrm{DPO}}^i$. We use quantile normalization to normalize the raw reward and obtain the final variance-reduced and numerically stable reward $\hat{r}_n^i$ (details in Appendix B).

While $\tilde{r}_n^i = -\mathcal{L}_{\mathrm{DPO}}^i$ might implicitly favor "easy" RMs that align with the current (potentially flawed) policy, this choice is justified by two factors:(i) Robustness to Overfitting: A "flattering" but low-quality RM (e.g., favoring short sequences) can only maintain low loss within a narrow distribution. As Thompson Sampling (TS) explores diverse RMs, the resulting policy updates quickly shift the distribution away from such biased regions, causing the flawed RM's loss to spike.(ii) Consistency as a Proxy for Quality: High-quality RMs typically exhibit self-consistent preference gradients, leading to training trajectories with lower variance and lower overall loss magnitude compared to inconsistent, low-quality RMs Shen et al. (2023); Liu et al. (2024a). By treating rewards as a time-series input to the Bayesian posterior, the router learns to favor RMs that provide long-term stable guidance rather than short-term agreement. Our empirical results further validate that this signal effectively identifies specialized RMs without collapsing to sub-optimal "yes-man" models.

### 3.4 OFFLINE–ONLINE INTEGRATION

While the offline and online routers can each perform RM routing as defined in Section 3.1, both have intrinsic limitations. The offline router leverages abundant supervised preference data but may degrade under distribution shift, whereas the online router adapts to the policy-induced distribution but suffers from cold start and exploration challenges. Thus, neither component alone is sufficient in practice.

To address this, our key idea is to combine their complementary advantages. A naïve approach is to directly combine their outputs (e.g., by weighted averaging of their predicted scores), but such schemes require a manually tuned global weight whose optimal value is unclear and may vary across training stages. Instead, we propose a more principled strategy based on prior injection. The insight is that both the offline BT head and the online Bayesian router can be viewed as linear models over the preference-pair embedding: the offline BT head computes $\langle \mathbf{h}, E_{\mathrm{bt}}[n] \rangle$ where $E_{\mathrm{bt}}[n]$ is the learned RM embedding, while the online router computes $\langle \mathbf{h}, \mathbf{w}_n \rangle$ where $\mathbf{w}_n$ is the latent RM weight vector. The semantic roles of $E_{\mathrm{bt}}[n]$ and $\mathbf{w}_n$ are thus closely aligned, differing mainly in the source of supervision (offline labels versus online rewards). This motivates initializing the online Bayesian router with the offline BT embeddings.

Concretely, we set the prior mean of each RM's weight vector in Eq. 4 to the corresponding offline embedding, i.e., $\boldsymbol{\mu}_n^{(0)} = E_{\mathrm{bt}}[n]$. This initialization provides the online router with prior knowledge about which types of preference pairs each RM is likely to handle well. As a result, it mitigates the cold-start problem and improves early routing accuracy. During training, the posterior distributions are iteratively refined using online rewards, allowing the router to adapt to the evolving policy-induced distribution while retaining the offline prior as a regularizer. In this way, BayesianRouter combines the robustness of offline training with the adaptivity of online learning.

## 4 EXPERIMENTS AND RESULTS

We evaluate BayesianRouter on instruction-following and reasoning benchmarks with the goal of demonstrating that it enables more effective reward model selection and consequently leads to superior alignment performance.

## 4.1 Experimental Setup

**Models.** We initialize the policy model with `LLaMA3-SFT-v2` released by (Dong et al., 2024). The reward model (RM) pool consists of $N = 4$ small yet high-performing models from the RewardBench 2 leaderboard (Malik et al., 2025): `Mistral-RM-for-RAFT-GSHF-v0` (RM_0), `GRM-Llama3.2-3B-rewardmodel-ft` (RM_1), `GRM-gemma2-2B-rewardmodel-ft` (RM_2), and `Skywork-Reward-V2-Qwen3-0.6B` (RM_3). For the offline router encoder, we use `SmolLM2-135M-Instruct` (see Appendix A.1).

**Datasets and Metrics.**

- **Offline preference datasets.** To train the offline router, we combine two human-annotated preference datasets: HelpSteer3 (Wang et al., 2025) and RM-Bench (Liu et al., 2024b), resulting in 50,402 preference pairs.
- **Instruction-following benchmarks.** Following (Dong et al., 2024), we evaluate the instruction-following ability on AlpacaEval-2 (Dubois et al., 2023), MT-Bench (Zheng et al., 2023), and Chat-Arena-Hard (Li et al., 2024). Policy models are trained on the `iterative-prompt-v1-iter3-20K` prompt set released by (Dong et al., 2024), and evaluated with length controlled AlpacaEval (Dubois et al., 2024), where model responses are compared against the SFT baseline using GPT-4 as the judge.
- **Reasoning benchmarks.** Following (Nguyen et al., 2024), we evaluate performance under different training distributions by training and testing on two reasoning benchmarks: GSM8K (Cobbe et al., 2021) and MMLU (Hendrycks et al., 2020). We report accuracy for both datasets, with detailed statistics provided in Appendix A.1.

**Baselines.** We compare BayesianRouter against the following methods:

- **Single RM:** Use a fixed RM from the pool for preference annotations.
- **Majority vote:** Annotate each preference pair with all RMs and select the final label via majority voting.
- **Random router:** Randomly select an RM to annotate a preference pair.
- **Uncertainty-Weighted Optimization (UWO)**: This ensemble method (Coste et al., 2023) downweights preference pairs that exhibit high disagreement among the RMs. We implement this by setting the weight for each pair to its consensus rate (i.e., the fraction of RMs that agree with the majority preference).
- **LASER:** The first RM routing method that employs LinUCB to select a single RM per batch (Nguyen et al., 2024).
- **w/o offline:** Variant of BayesianRouter without offline priors.
- **w/o online:** Variant of BayesianRouter that uses only the offline router for RM selection.

## 4.2 Main Results

Table 1 summarizes the performance of BayesianRouter against baseline methods. We have the following key observations: **(1)** BayesianRouter consistently outperforms all baselines on both instruction-following and reasoning benchmarks. Given that these results are achieved after training on datasets with distinct distributions, it demonstrates the adaptability of BayesianRouter to diverse types of training data. **(2)** BayesianRouter surpasses the performance of single RM baselines, showing that dynamically routing among multiple candidate RMs effectively aggregates their complementary strengths. In practice, users often choose a single RM based on leaderboard performance, which may not accurately reflect true RM performance across tasks or domains. Our results show that BayesianRouter eliminates this reliance and even surpasses the best-performing RM identified in hindsight. **(3)** BayesianRouter significantly outperforms the Majority Voting and UWO ensemble methods. While ensemble methods can also leverage complementary RMs, they require $O(N)$ RM calls per query, making them impractical for scaling to large RM pools. In contrast, BayesianRouter achieves higher performance with only $O(1)$ RM calls. BayesianRouter also substantially outperforms the routing baselines Random Routing and LASER, further validating the effectiveness of our routing strategy. **(4)** BayesianRouter outperforms its two ablations, *w/o offline* and *w/o online*, highlighting the complementary contributions of the offline-learned prior and the online Bayesian feedback loop. Removing either component leads to a notable degradation.

Notably, the *w/o offline* variant exceeds LASER, showing that our per-query Bayesian Thompson sampling router is superior to LASER's per-batch LinUCB approach.

Table 1: Main results on instruction-following and reasoning benchmarks.

| Method | Instruction-Following | | | Reasoning | |
|---|---|---|---|---|---|
| | AlpacaEval-2 | MT-Bench | Chat-Arena-Hard | GSM8K | MMLU |
| SFT | 50.00 | 50.00 | 50.00 | 67.63 | 54.29 |
| RM0 | 56.02 | 52.50 | 59.60 | 72.78 | 56.00 |
| RM1 | 61.86 | 56.25 | 64.80 | 74.22 | 57.03 |
| RM2 | 59.50 | 53.75 | 63.20 | 73.92 | 56.57 |
| RM3 | 60.37 | 52.50 | 62.00 | 74.53 | 56.28 |
| Majority vote | 60.75 | 53.75 | 63.40 | 74.22 | 56.71 |
| Random router | 58.39 | 52.50 | 61.20 | 73.46 | 56.07 |
| UWO | 61.74 | 56.25 | 63.60 | 74.30 | 56.43 |
| LASER | 60.50 | 51.25 | 62.40 | 74.00 | 56.35 |
| w/o offline | 60.99 | 53.75 | 63.20 | 74.37 | 56.64 |
| w/o online | 61.61 | 57.50 | 64.40 | 74.68 | 56.85 |
| BayesianRouter | **63.23** | **58.75** | **66.20** | **75.66** | **57.39** |

## 4.3 ADDITIONAL ANALYSIS OF BayesianRouter

**Effectiveness of Offline Router** We evaluate the offline router's ability to route preference pairs to the most suitable RM and analyze the factors affecting its performance. For in-distribution (ID) evaluation, we use the official test split of the HelpSteer3 dataset; for out-of-distribution (OOD) evaluation, we adopt RewardBench 2. Each prompt in RewardBench 2 is paired with one preferred response and three rejected responses, which we flatten into chosen–rejected pairs, discarding all ties. We further filter both test sets to retain only those samples where at least one candidate RM correctly identifies the preferred response, yielding 1,723 ID and 2,939 OOD preference pairs. Under this setup, an oracle router achieves 100% accuracy. To better understand the router's behavior, we introduce two additional baselines: *w/o CLS*, which removes the classification head from the offline router, and *0.5B encoder*, which replaces the SmolLM2-135M-Instruct encoder with Qwen2.5-0.5B-Instruct. Table 2 reports the results. Overall, our offline router substantially outperforms single-RM, majority voting, and random routing baselines in the ID setting, and also delivers consistent gains in the OOD setting, though with smaller margins. Nevertheless, there remains a considerable gap from the oracle, highlighting both the effectiveness and the generalization challenges of offline routing—likely due to limitations in the scale, diversity, or domain coverage of available preference data. This underscores the importance of BayesianRouter's online adaptation. In addition, removing the classification head leads to performance degradation, validating the benefit of multi-objective training. Finally, while larger encoders yield modest improvements in routing accuracy, we adopt the 135M encoder as a practical balance between performance and efficiency.

**Controlled simulation of online DPO.** In practical online DPO training, it is infeasible to obtain real-time human annotations for policy-generated responses, making it impossible to directly verify whether the RM selected by a router produces correct preference labels. To address this, we design a controlled simulation using the 2,939 human-labeled preference pairs from RewardBench 2. Instead of sampling responses from a live policy, we replay existing pairs and let the router select an RM to label them. Since ground-truth labels are available, we can measure how often the chosen RM provides the correct annotation, thereby directly assessing the quality of routing. Table 3 compares BayesianRouter with its ablations. The results show that BayesianRouter achieves the highest annotation accuracy during training and attains the best downstream alignment performance. This confirms that BayesianRouter's gains originate from more accurate RM routing rather than other confounding factors. The overall decrease in performance compared to the main results is attributed to the limited number of training samples. See more results in Appendix C.

**Ablation on integration strategy.** To validate the effectiveness of combining the offline and online routers, we compare BayesianRouter with a simple variant, *Weighted-score*. For each prefer-

Table 2: In-distribution and Out-of-distribution performance comparison.

| Method | In-distribution | Out-of-distribution | | | | | |
|---|---|---|---|---|---|---|---|
| | Score | Factuality | Pre IF | Math | Safety | Focus | All |
| RM_0 | 77.54 | 75.44 | 59.22 | 78.76 | 85.10 | 75.61 | 77.61 |
| RM_1 | 81.43 | 84.33 | 68.44 | 81.18 | 96.00 | **95.73** | 87.65 |
| RM_2 | 79.51 | 80.04 | 67.38 | 79.03 | **97.60** | 90.85 | 85.88 |
| RM_3 | 81.14 | 77.64 | **70.92** | **90.05** | 92.70 | 92.38 | 85.34 |
| Majority | 83.17 | 77.74 | 67.73 | 85.75 | 96.50 | 90.85 | 85.64 |
| Random | 79.80 | 79.94 | 65.96 | 82.80 | 92.80 | 87.80 | 84.48 |
| **Ours** (w/o CLS) | 89.73 | 84.12 | 66.67 | 85.22 | 96.20 | 92.99 | 87.34 |
| **Ours** (135M) | 90.31 | 84.85 | 65.60 | 86.83 | 96.20 | 92.07 | 87.92 |
| **Ours** (0.5B) | **90.77** | **85.16** | 66.31 | 87.90 | 95.90 | 91.46 | **88.06** |

Table 3: Controlled online DPO results.

| Method | AlpacaEval-2 | MT-Bench | Chat-Arena-Hard | GSM8K | MMLU | Acc. |
|---|---|---|---|---|---|---|
| w/o offline | 55.78 | 54.84 | 55.87 | 67.78 | 54.61 | 85.68 |
| w/o online | 56.72 | 56.41 | 57.71 | 68.39 | 54.82 | 87.92 |
| **Ours** | **57.63** | **56.76** | **58.15** | **68.76** | **54.93** | **88.23** |

ence pair, *Weighted-score* computes two separate score vectors: $s_1$ from the offline router's Bradley–Terry head, representing each RM's estimated competence, and $s_2$ from the online router initialized with zero-mean priors, representing the RM's current reward estimates. To address scale differences, each score vector is converted into a probability distribution via softmax. The two distributions are then combined using a fixed weight $\alpha$ as $s = \alpha s_1 + (1-\alpha)s_2$, and the RM with the highest combined score is selected. We sweep $\alpha \in \{0.25, 0.5, 0.75\}$ and report the best-performing setting. Table 4 presents the results. BayesianRouter consistently outperforms the *weighted-score* variant across all datasets. This demonstrates that initializing the online router with offline BT embeddings provides a principled and more effective mechanism to integrate offline knowledge with online adaptation, rather than relying on a simple linear weighting scheme.

Table 4: Ablation studies on integration strategy.

| Method | AlpacaEval-2 | MT-Bench | Chat-Arena-Hard | GSM8K | MMLU |
|---|---|---|---|---|---|
| Weighted-score | 61.12 | 56.25 | 63.80 | 74.37 | 56.75 |
| **Ours** | **63.23** | **58.75** | **66.20** | **75.66** | **57.39** |

## 5 CONCLUSION

In this work, we addressed the problem of adaptive reward model (RM) selection in iterative DPO training pipelines. We proposed BayesianRouter, a hybrid framework that first learns a multi-task offline router to capture RM strengths from preference data, and then injects this prior knowledge into a Bayesian Thompson sampling–based online router. The resulting framework adaptively selects a single RM for each preference pair while continually refining its routing policy through online rewards. Extensive experiments show that BayesianRouter consistently surpasses single-RM methods, RM ensembles, and strong routing baselines, demonstrating its effectiveness. For future work, we plan to design routers that jointly optimize the trade-off between annotation accuracy and RM inference cost to further improve RLHF alignment under constrained computational budgets.

ACKNOWLEDGMENTS

This work was supported by the Ministry of Education of Singapore (MOE) Tier 1 grant A8003480.

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

# A EXPERIMENTS

## A.1 EXPERIMENTAL SETTINGS

**Training setup.**

- **Offline router training.** We adopt `SmolLM2-135M-Instruct` as the encoder of the offline router and perform full-parameter fine-tuning using AdamW with a learning rate of $2 \times 10^{-5}$, batch size 8, and $\lambda = 0.2$. We train for 2 epochs with weight decay 0.01.
- **Online policy training.** The policy model is initialized from `LLaMA3-SFT-v2`. We fine-tune it for 1 epoch using LoRA (rank 16, $\alpha = 32$) applied to the `q_proj` and `v_proj` projection matrices. We use a learning rate of $5 \times 10^{-6}$ with Adam. Each prompt batch contains 16 prompts; for each prompt we sample 6 candidate responses at temperature 0.8 and construct 4 preference pairs, yielding 64 preference pairs per batch. We set the parameter $\beta$ in Equation 1 to 1. We do not use prompt templates during training or inference. The maximum input length is 128 tokens and the maximum output length is 256 tokens, except for the second turn of MT-Bench, where the input length is increased to 512 to accommodate multi-turn context. When initializing the online router with offline-learned embeddings, we set the prior variance for the RM weights $\sigma_w^2 = 0.02$ (with noise variance $\sigma^2 = 1$), reflecting stronger trust in the prior; when using the online router alone, we set $\sigma_w^2 = 1$ with prior mean 0 and $\sigma^2 = 1$.

All experiments are conducted on a server equipped with $8\times$ RTX A6000 GPUs, each with 48GB memory.

**Benchmark Details.** For AlpacaEval-2, MT-Bench, and Chat-Arena-Hard, we use them exclusively as test sets. The details are as follows:

- AlpacaEval-2 (Dubois et al., 2023): This is a single-turn dialogue benchmark consisting of 805 prompts covering a wide range of topics.
- MT-Bench (Zheng et al., 2023): This is a multi-turn dialogue benchmark comprising 80 prompts across various domains. Each prompt contains two questions: the model first answers the initial question, then receives the initial question, its response, and the second question as input to produce a second response. The evaluation is conducted based on the quality of both responses jointly.
- Chat-Arena-Hard (Li et al., 2024): This benchmark contains 500 high-quality prompts selected from user queries in Chatbot Arena. It is designed to evaluate models on creativity, data analysis, deep comprehension, and problem-solving abilities.

For GSM8K and MMLU, we split each dataset into training and test sets. Policy models are fine-tuned on the training split and evaluated on the held-out test set. For GSM8K, we use `math_verify` to parse model responses and compute accuracy against the ground-truth labels. For MMLU, we use `xFinder` (Yu et al., 2025b) to extract the predicted choices before comparing them with the labels. Table 5 summarizes the number of instances in each split.

Table 5: Dataset statistics.

| Dataset | Train | Test |
|---------|-------|------|
| GSM8K | 7465 | 1319 |
| MMLU | 11233 | 2809 |

**Settings for Efficiency Analysis.** To evaluate the scalability of BayesianRouter, we extend the RM pool to 8 candidates. In addition to the 4 RMs used in the main experiments, we select 4 larger models from the RewardBench 2 leaderboard (Malik et al., 2025): `RISE-Judge-Qwen2.5-32B` (RM_4), `Skywork-Critic-Llama-3.1-8B` (RM_5), `Selene-1-Mini-Llama-3.1-8B` (RM_6), and `RISE-Judge-Qwen2.5-7B` (RM_7).

We measure training time on 8 GPUs. For BayesianRouter, Majority Voting, and LASER, we allocate resources as follows: RM_4 occupies 2 GPUs; RM_0, RM_5, RM_6, and RM_7 each occupy 1 GPU; RM_1, RM_2, and RM_3 share 1 GPU; and policy training uses the remaining GPU. For the single-RM baselines, we consider the slowest RM (RM_4) and the fastest RM (RM_3). In the slowest-RM setting, RM_4 is assigned 2 GPUs with data parallelism for policy training. In the fastest-RM setting, RM_3 occupies 1 GPU with data-parallel policy training.

## B   METHOD DETAILS

**MAB reward normalization.** To provide stable learning signals to the bandit router, we do not directly use raw per-pair losses as rewards. Instead, for each training step $t$ we first compute a batch-level baseline over all preference pairs in the batch:

$$\bar{\ell}_t \;=\; \frac{1}{|\mathcal{B}_t|} \sum_{i \in \mathcal{B}_t} \mathcal{L}^{(i)}(t),$$

where $\mathcal{B}_t$ denotes the set of preference pairs at step $t$ and $\mathcal{L}^{(i)}(t)$ is the training loss of pair $i$ under its selected RM. The instantaneous advantage-style reward for pair $i$ is then

$$r_i(t) \;=\; \bar{\ell}_t \;-\; \mathcal{L}^{(i)}(t),$$

which normalizes for batch difficulty and highlights the relative quality of each pair within the batch.

Because the scale of $r_i(t)$ may drift over time, following LASER (Nguyen et al., 2024), we further apply quantile-based rescaling. Let $\mathcal{R}_{1:t-1} \;=\; \{r_j(\tau) \mid \tau < t, j \in \mathcal{B}_\tau\}$ denote the set of past rewards up to step $t-1$. We compute the empirical $20^{\text{th}}$ and $80^{\text{th}}$ percentiles of this set, denoted $q_t^{lo}$

Table 6: Controlled online DPO results.

| Method | Instruction-Following | | | Reasoning | | Acc. |
|---|---|---|---|---|---|---|
| | AlpacaEval-2 | MT-Bench | Arena-Hard | GSM8K | MMLU | |
| SFT | 50.00 | 50.00 | 50.00 | 67.63 | 54.29 | - |
| RM0 | 53.05 | 53.33 | 53.39 | 67.17 | 54.18 | 77.61 |
| RM1 | 56.14 | 55.88 | 57.93 | 68.16 | 54.79 | 87.65 |
| RM2 | 55.83 | 54.55 | 56.32 | 67.70 | 54.54 | 85.88 |
| RM3 | 55.28 | 54.05 | 55.60 | 67.55 | 54.33 | 85.34 |
| Majority | 55.66 | 54.29 | 56.92 | 67.40 | 54.47 | 85.64 |
| Random | 53.55 | 53.13 | 54.51 | 67.25 | 54.18 | 84.48 |
| UWO | 56.19 | 55.56 | 56.65 | 67.78 | 54.61 | 85.64 |
| LASER | 55.20 | 55.17 | 54.96 | 67.40 | 54.40 | 85.54 |
| w/o off. | 55.78 | 54.84 | 55.87 | 67.78 | 54.61 | 85.68 |
| w/o on. | 56.72 | 56.41 | 57.71 | 68.39 | 54.82 | 87.92 |
| BayesianRouter | **57.63** | **56.76** | **58.15** | **68.76** | **54.93** | **88.23** |

and $q_t^{hi}$. The normalized reward is then

$$\hat{r}_i(t) = \begin{cases} 0 & \text{if } r_i(t) < q_t^{lo}, \\ 1 & \text{if } r_i(t) > q_t^{hi}, \\ \dfrac{r_i(t) - q_t^{lo}}{q_t^{hi} - q_t^{lo}} & \text{otherwise.} \end{cases}$$

This two-stage procedure—batch-baseline centering followed by quantile scaling—yields rewards that are both variance-reduced and numerically stable across the training process.

## C  ADDITIONAL EMPIRICAL RESULTS

**Controlled online DPO results.**  Table 6 presents the complete controlled online DPO results, using the same training setup as described in Appendix A.1.

**Analysis of reward design.**  We further analyze the effect of different reward formulations for online routing. In particular, we design two alternative variants that compute rewards based on RM-to-RM comparisons rather than batch-level normalization.

*Full_Advantage.*  For each preference pair, all candidate RMs are queried to obtain their induced training losses. The router still selects a single RM via Thompson sampling, but the reward is defined as a binary advantage: if the selected RM's loss is no greater than the average loss across all RMs, the reward is set to 1, otherwise to 0. This design removes the confounding effect of sample difficulty and purely reflects the relative quality of RMs.

*Light_advantage.*  As a more scalable compromise, we randomly sample $C = 3$ RMs to compute the baseline average, instead of evaluating the full pool. This reduces computational overhead while still approximating the comparative signal.

We compare these two variants against our proposed batch-normalized reward (with quantile rescaling) under the controlled online DPO setup on RewardBench 2. Table 7 reports the results. As expected, the *Full_advantage* variant achieves the strongest performance, since it leverages the most informative RM comparisons, but at the cost of prohibitive computation and poor scalability. The *Light_advantage* variant attains performance close to the Full_advantage while being more efficient, showing that subsampling can preserve much of the benefit. Our proposed batch-normalized reward is less competitive in isolation, but it is by far the most efficient and, when combined with the offline prior, yields the best overall alignment performance.

**Empirical Justification for the Linear Head**  A potential concern regarding our architecture is whether a linear model is sufficiently expressive to capture the complexities of reward model (RM)

Table 7: Ablation studies on reward design.

| Method | AlpacaEval-2 | MT-Bench | Arena-Hard | GSM8K | MMLU | Acc. |
|---|---|---|---|---|---|---|
| Full_advantage | **56.50** | **56.52** | **57.51** | **68.46** | **54.79** | **87.72** |
| Light_advantage | 56.12 | 55.81 | 56.13 | 67.85 | 54.72 | 87.38 |
| **Ours** (w/o offline) | 55.78 | 54.84 | 55.87 | 67.78 | 54.61 | 85.68 |

selection, which might involve high-order feature interactions. We argue that since our router operates on embeddings produced by a task-aligned LLM-based encoder, the non-linear semantic structures are already internalized within the representation space. To empirically validate this, we conducted a diagnostic study in the offline setting. We froze the pre-trained encoder and compared the performance of two different prediction heads: (i) a Linear head, utilizing a standard Bradley-Terry (BT) regression layer; (ii) a Non-linear head, consisting of a single-layer MLP (with ReLU activation) followed by the BT layer.

As shown in Table 8, the non-linear head yields only marginal improvements over the linear head in both In-Distribution (ID) and Out-of-Distribution (OOD) scenarios. The fact that a significantly more expressive non-linear head fails to provide substantial gains suggests that the encoder successfully organizes the preference-pair embeddings into a largely linearly separable space for the RM-selection task. Consequently, a linear head is not an oversimplification but a sufficient and computationally efficient choice for both offline prediction and online Bayesian updates.

Table 8: Comparison of Linear and Non-linear (MLP) prediction heads on top of frozen LLM embeddings. Results are averaged over 3 independent runs.

| Output Head | ID Accuracy (%) | OOD Accuracy (%) |
|---|---|---|
| MLP + BT head | $91.41 \pm 0.10$ | $88.42 \pm 0.06$ |
| Linear BT head (Ours) | $91.31 \pm 0.09$ | $88.35 \pm 0.04$ |

## C.1 SUFFICIENCY OF UNIMODAL POSTERIOR UNDER DISTRIBUTION SHIFT

A critical assumption in our Bayesian Linear Thompson Sampling (BLTS) is that the uncertainty in RM selection can be adequately modeled by a unimodal Gaussian posterior. One might wonder if the shifting policy distribution during online training could induce complex, multi-modal uncertainty that a single Gaussian cannot capture. To investigate this, we compared BLTS against a more expressive Online Bootstrap Thompson Sampling variant. This baseline maintains $M = 10$ independent linear heads; at each step, it randomly selects one head for routing and updates each head using Poisson(1) resampling weights to approximate a non-parametric posterior. This approach is generally more robust in capturing multi-modal or complex reward distributions.

As shown in Table 9, BLTS consistently outperforms the Bootstrap TS variant across both reasoning (GSM8K) and general knowledge (MMLU) benchmarks. The observation that a more flexible posterior family offers no additional gain—and even slightly underperforms—suggests that the task-aligned encoder successfully maps preference pairs into a representation space where the uncertainty remains well-behaved. We attribute this to the high-quality and diverse preference data used during the offline pre-training phase, which allows the encoder to collapse complex multi-modal semantic structures into a stable embedding space that remains effective even as the policy evolves.

Table 9: Comparison between Online Bootstrap Thompson Sampling and our Bayesian Linear Thompson Sampling (BLTS).

| Method | GSM8K | MMLU |
|---|---|---|
| Bootstrap TS ($M = 10$) | 74.15 | 56.43 |
| BLTS (Ours) | 74.37 | 56.64 |

## C.2 TRAINING EFFICIENCY.

We evaluate the training efficiency of BayesianRouter. Unlike majority-voting ensembles that require $O(N)$ RM calls per preference pair, BayesianRouter selects a single RM at $O(1)$ cost. The extra overhead relative to a single-RM baseline comes from (i) encoding preference pairs with the offline router and (ii) lightweight Bayesian posterior updates. Both are independent of the number and size of candidate RMs and amortize as the pool grows. Moreover, if the router often selects smaller, cheaper RMs, the overall cost may even fall below using a single large RM. To demonstrate the scalability of Bayesian-Router, we increase both the size and number

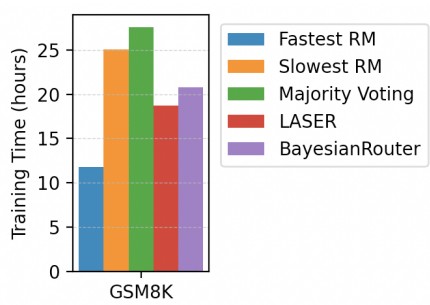

Figure 2: Training efficiency.

of candidate RMs and compare wall-clock training time on the GSM8K dataset against four baselines: the fastest single RM, the slowest single RM, Majority Voting, and LASER. Figure 2 shows that while BayesianRouter is slower than the fastest single RM, it substantially outperforms Majority Voting and the slowest single RM, demonstrating its efficiency. LASER runs marginally faster than BayesianRouter because it reuses policy embeddings rather than encoding preference pairs independently. Detailed experimental settings are provided in Appendix A.1.

# D  LLM USAGE

We acknowledge the limited use of large language models (LLMs) as an editorial assistant to enhance the clarity, grammar, and overall linguistic quality of our manuscript. Specifically, an LLM was employed for minor stylistic improvements and grammatical corrections, as well as to refine sentence structures for better readability. No LLM was used for content generation, ideation, methodological design, experimental execution, or data analysis. All scientific content, intellectual contributions, and experimental results presented in this paper are solely the work of the human authors. The authors take full responsibility for the entirety of the paper's content, including any text that may have been subjected to LLM-assisted polishing.

