# OpenReview forum: "Reward Model Routing in Alignment"
_ICLR.cc/2026/Conference — ICLR 2026 Poster_

### Official Review · Reviewer_PxRA · 2025-10-25

**Soundness:** 3
**Presentation:** 3
**Contribution:** 3
**Rating:** 4
**Confidence:** 4

**Summary:**

The authors propose a Bayesian approach to online reward model selection to improve LLM performance.
The approach combines offline and online data. With offline data, a neural network representation of preference pairs alongside two prediction heads for prompt-response preference pairs is trained. Based upon this model, the reward model with best prediction performance for a given prompt-response preference pair is selected.   With online data, the authors use a Bayesian linear regression model of reward model "utility" (based upon the representation obtained offline). A sample of reward models from the posterior distribution is used to select the model better suited for a given a prompt-response preference pair (Thompson-sampling).
The proposed approach is evaluated on experimental testbed with alternative reward model selection models such as majority opinion, uncertainty-weighted optimization and linear upper confidence bound (as in Nguyen et al., 2024).

**Strengths:**

1. The proposed approach aims to provide the RLHF training paradigm with the ability to adapt according to changing tasks while minimizing additional computational effort.

2. The numerical results indicate the proposed that reward model selection mechanism outperforms alternative approaches for model selection.

**Weaknesses:**

1. The choice of a linear model for utility is presented in ad-hoc fashion. A linear model has significant limitations for example it fails to capture interactions between features (e.g., "helpfulness is only valued if tone is polite") or hierarchical semantics (e.g., "if response is correct, then prefer shorter length; otherwise, correctness dominates").

2. The performance of the linear model of utility relies heavily on the representation quality (which is obtained offline).  Any attempt to jointly re-train representation and utility will take away the simplicity of Bayesian updating of a linear-gaussian model.

3. The choice of Gaussian noise is also ad-hoc. One can not represent multimodal preference beliefs (e.g., two equally plausible user preference patterns) with such model.

**Questions:**

1. I encourage the authors to provide arguments to support ad-hoc modeling choices (linear model of utility, robustness of neural representation for different tasks, Gaussian noise).

2. I find the alternative model selection models may not provide a competitive benchmark. Perhaps the linear upper confidence bound proposed in Nguyen et al., 2024 duly extended to account for preference pairs might be a more decent comparison.

2. Thompson sampling optimizes immediate expected gain under a single posterior sample. This can lead to suboptimal long-term exploration (over-exploitation) and inefficient learning when preferences are high-dimensional or multimodal.

---

> ### Author Response · Authors · 2025-11-20
>
> Dear Reviewer PxRA,
> We would like to thank you for your detailed questions and feedback. We respond point by point below.
>
> > [W1] The choice of a linear model for utility is presented in ad-hoc fashion. A linear model has significant limitations for example it fails to capture interactions between features (e.g., "helpfulness is only valued if tone is polite") or hierarchical semantics...
> >
> > [Q1] I encourage the authors to provide arguments to support ad-hoc modeling choices...
>
> **The use of a linear model for utility estimation is not an ad-hoc simplification but a principled architectural choice, and it does not suffer from the inability to represent feature interactions or hierarchical semantics.** The linear model is never applied directly to the raw semantic space; it operates on the task-aligned embedding learned by the offline LLM-based encoder. This encoder is built with an LLM backbone and a single-layer MLP, and is trained offline specifically for RM selection, where the linear head is optimized to identify the best RM. This supervised offline training shapes the embedding space into a low–effective-dimensional representation that is largely linearly separable for the RM-selection task. In other words, **non-linear structure—feature interactions and hierarchical semantics—is handled by the encoder**, while the online router becomes a deep contextual bandit with a non-linear representation and a Bayesian linear head, a well-studied and widely validated architecture in contextual bandit literature [2].
>
> Our empirical results support this design: Table 2 shows that the learned embedding enables accurate RM selection with a linear head in the offline setting, and Table 1 demonstrates that linear Thompson Sampling on the same representation produces effective RM routing online. Similar architectural choices have also proven effective in prior work: for offline LLM selection, architectures such as P2L[1] also use LLM-based non-linear feature extractors with a linear prediction head and report state-of-the-art results. For online selection, deep representation + linear Thompson sampling is a widely used and theoretically grounded approach in contextual bandits[2].
>
> > [W2] The performance of the linear model of utility relies heavily on the representation quality (which is obtained offline). Any attempt to jointly re-train representation and utility will take away the simplicity of Bayesian updating of a linear-gaussian model.
>
> We totally agree that "The performance of the linear model of utility relies heavily on the representation quality (which is obtained offline)", and we do not attempt to jointly re-train the representation and the utility. However, **this observation does not constitute a limitation of our method**, for the following reasons:
>
> **First, the offline representation is already highly task-aligned and sufficient.** High-quality fixed representations combined with linear contextual bandits are widely adopted and empirically validated as an effective design choice [2]. Our context encoder uses an LLM backbone with a single-layer MLP and is trained offline specifically for RM selection. The linear head is optimized to identify the best RM for each preference pair, and this objective is fully aligned with the online utility function used by the router. Consequently, the offline training shapes a low–effective-dimensional, largely linearly separable embedding space tailored for RM selection. Given such task-aligned representations, a linear Thompson sampler is entirely adequate for modeling RM utilities. The empirical evidence in Table 1 and Table 2 clearly supports this. We believe that even larger and more diverse offline preference datasets could further improve representation quality, but this does not affect the core contribution of our work.
>
> **Second, online encoder re-training is impractical and orthogonal to our design goals.** Because the encoder contains an LLM backbone, any online update—even with LoRA—would incur prohibitive computational overhead if performed at every RLHF step. Besides, updating the encoder would fundamentally break the conjugacy assumptions required for closed-form Bayesian updates in a linear–Gaussian Thompson sampler. Although some contextual-bandit works explore periodic representation updates [3], such approaches remain expensive and may even underperform a frozen, well-trained representation due to limited adaptation. **Thus, avoiding online encoder updates is both deliberate and necessary in our setting, and does not reflect a methodological weakness.**
>
> [1] Prompt-to-Leaderboard, 2025 arXiv, https://arxiv.org/pdf/2502.14855
>
> [2] Leveraging Good Representations in Linear Contextual Bandits, 2021 ICML, https://arxiv.org/pdf/2104.03781
>
> [3] Deep Bayesian Bandits Showdown: An Empirical Comparison of Bayesian Deep Networks for Thompson Sampling, 2018 ICLR,  https://arxiv.org/pdf/1802.09127

---

> > ### Comment · Reviewer_PxRA · 2025-11-20
> > **Response**
> >
> > Thank you for the clarification. The empirical results in Tables 1–2 clearly show that the current architecture performs well; my concern is only that this does not on its own demonstrate that the encoder truly renders RM selection linearly separable. A few small checks—such as an ablation using a shallow nonlinear head, a linear-probe diagnostic on the embeddings, or a drift sensitivity test across policy iterations—would make the justification for the linear head much more convincing and help distinguish genuine linear separability from performance that is simply “good enough” for these datasets.

---

> > > ### Author Response · Authors · 2025-11-23
> > >
> > > > my concern is only that this does not on its own demonstrate that the encoder truly renders RM selection linearly separable.  A few small checks—such as an ablation using a shallow nonlinear head, a linear-probe diagnostic on the embeddings, or a drift sensitivity test across policy iterations—would make the justification for the linear head much more convincing...
> > >
> > > Thank you for the insightful comment. We agree that additional diagnostics can strengthen the justification for using a linear prediction head. Following your suggestion, we conducted an additional diagnostic study: we freeze the offline router’s encoder and compare two heads trained on top of the same embeddings—(i) a non-linear head consisting of a single-layer MLP followed by the BT regression head, and (ii) a linear BT regression head. All hyperparameters are kept identical, with the only difference being the extra MLP layer. The results are shown below (run 3 times to obtain the mean and standard deviation). We observe that the non-linear variant improves over the linear head only marginally. Moreover, the non-linear head contains more parameters, meaning it benefits from a larger model capacity. The fact that it still yields only marginal gains, despite having more parameters, suggests that additional non-linearity is not providing substantial benefits on top of the encoder’s representation. Taken together, this supports that the encoder already organizes the representation space such that RM selection is effectively linearly separable, making a linear BT head sufficient. We will incorporate this experiment and its discussion into the revised paper.
> > >
> > > | Output layer  | ID Accuracy (%)      | OOD Accuracy (%)     |
> > > |---------------|-----------------------|------------------------|
> > > | MLP + BT head | 91.41 ± 0.10         | 88.42 ± 0.06          |
> > > | BT head       | 91.31 ± 0.09         | 88.35 ± 0.04          |

---

> ### Author Response · Authors · 2025-11-20
>
> > [W3] The choice of Gaussian noise is also ad-hoc. One can not represent multimodal preference beliefs (e.g., two equally plausible user preference patterns) with such model.
>
> **The choice of Gaussian residuals is not an ad-hoc hack but the modeling assumption required by Bayesian linear Thompson Sampling (BLTS) [1]**: BLTS models rewards as a linear map over context embeddings plus Gaussian noise, which yields a closed-form Gaussian posterior and enables efficient posterior sampling. The reviewer’s claim that this prevents representing “multimodal preference beliefs” can be read in two ways. One interpretation is that an individual RM’s reward surface is multimodal (e.g., an RM performs well on both safety and math contexts); the other is that, for a given context embedding, there may be two mutually plausible explanatory modes that point to different best-arms (e.g., the embedding combines math and creative signals and RM1 suits the math interpretation while RM1 suits the creative interpretation).
>
> **Neither interpretation undermines our BLTS router in practice.** For the first case, we operate on a task-specialized embedding learned offline (LLM backbone + single-layer MLP) with supervised training for RM selection: this encoder nonlinearly folds complex utility structure into a low–effective-dimensional representation that is largely linearly separable for the RM-selection objective, so multimodal reward regions are absorbed by the embedding and a linear posterior over that embedding is sufficient. For the second case, BLTS naturally handles situations where multiple arms are plausibly optimal [1]: even with a unimodal Gaussian posterior over parameters, sampling in weight space induces appropriate selection probabilities across such arms. Returning to the two examples above, the math vs. safety reward regions for RM0 are folded by the embedding, and the combined math+creative context allows both RM1 and RM2 to receive substantial selection probability. Empirically, Table 1 further confirms that BLTS effectively models RM utilities in our setting.
>
> [1] A Tutorial on Thompson Sampling, 2017 arXiv, https://arxiv.org/pdf/1707.02038
>
> > [Q2] I find the alternative model selection models may not provide a competitive benchmark. Perhaps the linear upper confidence bound proposed in Nguyen et al., 2024 duly extended to account for preference pairs might be a more decent comparison.
>
> We respectfully disagree with the claim that our model-selection baselines are not competitive. **We have made a deliberate effort to include all strong and relevant baselines available in the literature.** The reviewer’s suggestion corresponds to enhancing LASER using a design proposed by our paper, namely replacing question-only context with preference-pair context. **While this is an interesting idea, modifying a prior method with our own concrete design is generally not considered a standard or fair baseline practice in the community.**
>
> **Coverage of strong baselines.** To the best of our knowledge, our method is the only online+offline RM router in the literature. LASER is the only existing online RM router and is included. Although no prior work provides an offline-only RM router, we adapt the SOTA LLM-routing method P2L (2025)—a Bradley–Terry–based approach—into the w/o CLS variant in Table 2. It performs substantially worse than our multi-task offline router. We further include SOTA RM-ensemble methods (e.g., majority voting). Altogether, our baseline suite covers all competitive and practically relevant alternatives we are aware of.
>
> **Exceeding the single-best RM is non-trivial and highly meaningful.**
> In practice, users typically rely on RM leaderboards to select a single RM for RLHF training. However, RM rankings often invert across benchmarks. The “single-best” baseline is therefore a hindsight oracle, not attainable in practice. As noted in prior LLM-routing literature, only P2L is known to surpass single-best on OOD datasets; most routing methods fail to do so. Our method consistently outperforms all strong baselines—including single-best RM—across all benchmarks, underscoring its practical and scientific value.
>
> **On the suggested “extended LASER” variant.** We appreciate the reviewer’s idea of extending LASER by replacing question-only context with preference-pair context. We implemented precisely this variant on top of the original LASER pipeline while keeping all other design choices unchanged. The results (see Table below) show no meaningful improvement over the original LASER. This may be because the LinUCB component used in LASER tends to under-explore. Our Bayesian online router substantially outperforms both the original and extended LASER variants, so all conclusions in the paper remain unchanged.
>
> | Method        | GSM8K | MMLU |
> |-------------------|----------:|----------:|
> | LASER             | 74.00     | 56.35     |
> | LASER enhanced    | 74.22     | 56.46     |
> | BayesianRouter | 75.66 | 57.39|

---

> > ### Comment · Reviewer_PxRA · 2025-11-20
> > **Response**
> >
> > Thank you for the clarification. I agree that Gaussian noise is standard in BLTS, but my concern is about whether a unimodal Gaussian posterior is a good approximation of the uncertainty structure in RM selection as the policy distribution shifts. The current results show good point prediction, but do not demonstrate that multimodal or task-dependent uncertainty is actually collapsed by the encoder. A few small checks—such as posterior-calibration diagnostics, a comparison to a bootstrap/ensemble TS variant, or a stress test on mixed-task embeddings—would help validate that a single Gaussian posterior is indeed sufficient in this setting.

---

> ### Author Response · Authors · 2025-11-20
>
> > [Q3] Thompson sampling optimizes immediate expected gain under a single posterior sample. This can lead to suboptimal long-term exploration (over-exploitation) and inefficient learning when preferences are high-dimensional or multimodal.
>
> Our online router does not operate in a generic high-dimensional semantic embedding space. The preference-pair encoder used online is the LLM+single-layer MLP encoder trained offline specifically for RM selection, where a linear head is optimized to identify the best RM. This supervision shapes the encoder into a task-aligned, low-effective-dimension representation in which RM-relevant distinctions are largely linearly separable and the multimodal structure of raw semantic space is already collapsed by the encoder (Table 2 demonstrates that a linear model can accurately predict RM choices in this embedding space.). **Consequently, the online Bayesian linear Thompson sampler is applied not to a complex, multimodal reward landscape, but to an embedding space intentionally constructed to support linear discrimination.** Under this representation, the concerns about high-dimensional preference structure and insufficient long-term exploration no longer apply, and thus it can be effectively handled by linear Thompson sampling. The experimental results in Table 1 also demonstrate that the online router achieves highly competitive performance, proving its effectiveness in exploration and learning.
>
> We sincerely thank the reviewer again for the insightful comments, which have helped us significantly improve the clarity and completeness of the paper. We hope that our responses above adequately address the reviewer’s concerns, and we would be more than happy to further discuss any remaining questions.
> Finally, we would like to emphasize the main contribution of our work: a novel offline+online RM routing framework that injects offline RM embeddings as a principled Bayesian prior into an online bandit router. This hybrid design has not appeared in prior routing literature, and our experiments consistently demonstrate the effectiveness and practical value of this idea across multiple benchmarks. We hope that, with these clarifications, the reviewer will recognize the novelty and significance of the contribution and revisit the score.

---

> ### Author Response · Authors · 2025-11-23
>
> > I agree that Gaussian noise is standard in BLTS, but my concern is about whether a unimodal Gaussian posterior is a good approximation of the uncertainty structure in RM selection as the policy distribution shifts...
>
> Thank you for the insightful comment. We agree that additional diagnostics can help assess whether the encoder maintains a representation where a unimodal posterior remains adequate as the policy distribution shifts. Following your suggestion, we implemented a standard Online Bootstrap Thompson Sampling variant with M=10 bootstrap linear models. At each round, we randomly sample one bootstrap head to select the RM, collect the reward, and update each head according to its Poisson(1) resampling weight. As shown below, BLTS outperforms bootstrap TS on both GSM8K and MMLU. This indicates that allowing a more expressive posterior family offers no additional gains, suggesting that the encoder indeed maps preference pairs into an embedding space where RM-selection uncertainty is well-captured by a single Gaussian posterior. We will include this experiment and its analysis in the revised paper.
>
> | Method  | GSM8K     | MMLU     |
> |---------------|-----------------------|------------------------|
> | bootstrap TS | 74.15         | 56.43          |
> | BLTS       | 74.37         | 56.64          |
>
> We hope that the additional evidence satisfactorily resolves your concerns. We would be glad to further clarify any remaining questions. Thank you again for your constructive feedback and for reconsidering our submission in light of these results.

---

> > ### Comment · Reviewer_PxRA · 2025-11-23
> > **Response**
> >
> > I have updated my numerical evaluation to reflect the discussion.

---

> > > ### Author Response · Authors · 2025-11-24
> > >
> > > Dear PxRA, thank you very much for your constructive feedback and for considering our responses in your numerical evaluation. I noticed that the evaluation score has not changed on my side, so I just wanted to check if the updated evaluation has been posted. If you have any remaining concerns, we would be happy to continue the discussion. Thanks again for your thoughtful comments.

---

### Official Review · Reviewer_xgBK · 2025-10-25

**Soundness:** 3
**Presentation:** 3
**Contribution:** 2
**Rating:** 6
**Confidence:** 3

**Summary:**

The paper proposes BayesianRouter, a hybrid reward-model (RM) routing framework for preference-based alignment (DPO family). Experiments on instruction following (AlpacaEval-2, MT-Bench, Arena-Hard) and reasoning (GSM8K, MMLU) claim consistent gains over: best single RM, ensembling (majority, UWO), and the prior routing baseline LASER

**Strengths:**

1. Clear problem framing: RM routing is a compelling middle ground between costly O(N) ensembles and single-RM training.
2. Using offline BT embeddings as priors for a Bayesian linear bandit seems a pretty interesting design.
3. Main results across five benchmarks show better performance over existing works
4. The paper argues O(1) RM calls with lightweight extra overhead, and compares wall-clock training to single-RM, LASER, and majority vote

**Weaknesses:**

1. To my understanding, the main novelty of this work is the prior injection from offline embeddings into online TS. The contribution is solid engineering + good fit to RM routing, but may feel unsurprising conceptually.
2. The online update uses (normalized) negative DPO loss per routed pair as the reward. This couples router training tightly to the current policy’s loss landscape and may conflate pair difficulty and RM quality.
3. Concerns on evaluation choices & fairness (see questions for details)

**Questions:**

1. Judge LLM and length-controlled AlpacaEval are used for instruction following evaluation. Would this risks judge-model bias? Results should report variance across judges.
2. Were their hyperparameters tuned per RM ($\beta$, LoRA rank, sampling temperature), or held constant—potentially under-optimizing some RMs?
3. For the computation cost, the router adds encodings + bandit updates; wall-clock plots are described, but exact GPU hours and per-step RM latency distributions (small vs large RMs) would clarify cost/quality trade-offs.
4. Offline router shows a non-trivial gap in OOD case; the paper argues online adaptation bridges it, but I don’t see stress tests where online distribution shifts during training. Is it possible to include such an experiment study?
5. Since reward is a function of policy loss, how stable is routing across training (do posteriors collapse to 1–2 RMs)? Please include posterior traces $(\mu,\Sigma)$ over steps and the share of traffic per RM if possible.
6. With larger pools (>8 RMs), do we still get O(1) advantages in practice? Any diminishing returns or exploration debt? (You mention scaling, but numbers are brief.)

Overall I think the paper tackles a real pain point in RLHF/RLAIF with a clean method and interesting results. Addressing my above concerns could further strengthen the paper.

---

> ### Author Response · Authors · 2025-11-21
>
> Dear xgBK,
>
> Thank you for your insightful comment. We respond point by point below.
>
> > [W1] To my understanding, the main novelty of this work is the prior injection from offline embeddings into online TS. The contribution is solid engineering + good fit to RM routing, but may feel unsurprising conceptually.
>
> We would like to say the contribution was underestimated. Our work is not merely “injecting an offline prior into TS.” It introduces: (1) the first offline + online hybrid paradigm for RM routing—conceptually unsurprising in hindsight, but not obvious, as no prior RM or LLM routing method employs the offline prior + online adaptation paradigm. (2) a novel multi-task offline router (BT + CLS) specifically designed to model RM specialization, which is new to both RM routing and LLM routing; and (3) a principled, non-trivial prior‐injection mechanism that leverages the structural alignment between offline RM embeddings and Bayesian TS posteriors. Empirically, this design clearly outperforms naive score fusion (Table 4).
>
> > [W2] The online update uses (normalized) negative DPO loss per routed pair as the reward. This couples router training tightly to the current policy’s loss landscape and may conflate pair difficulty and RM quality.
>
> We would like to say that the normalized negative DPO loss is already a reasonable and effective bandit reward, sufficient to enable BayesianRouter to outperform strong baselines across diverse benchmarks. Designing a more theoretically “perfect” reward is not the focus or contribution of this work, and we leave it for future research.
>
> First, high-quality RMs induce long-run DPO-loss trajectories with lower variance and lower overall magnitude than low-quality RMs due to their more self-consistent preference gradients [1][2], and therefore accumulate a long-term posterior advantage. Second, to reduce difficulty-induced noise, we do not use raw losses; instead we apply (i) batch-based advantage normalization to alleviate absolute pair difficulty, and (ii) quantile rescaling to stabilize reward dynamics over time. Third, our empirical results demonstrate that BayesianRouter consistently selects higher-quality RMs and achieves state-of-the-art downstream performance. This confirms that the normalized negative DPO-loss reward is sufficiently reliable for practical RM routing in online DPO.
>
> [1] The Trickle-down Impact of Reward Inconsistency on RLHF, ICLR 2024, https://openreview.net/pdf?id=MeHmwCDifc
>
> [2] Elephant in the Room: Unveiling the Impact of Reward Model Quality in Alignment, 2024 arXiv, https://arxiv.org/pdf/2409.19024
>
> > [Q1] Judge LLM and length-controlled AlpacaEval are used for instruction following evaluation. Would this risks judge-model bias? Results should report variance across judges.
>
> We agree that the choice of LLM judges may influence fairness and accuracy, and we will include aggregated results from multiple judges in the paper as suggested. However, we would also like to say that using a single strong LLM as the judge is a widely adopted evaluation practice in the community [1][2]. Moreover, GSM8K and MMLU have standardized ground-truth answers and thus are not affected by judge-model bias, providing reliable evidence for the effectiveness of our method.
>
> [1] Direct Language Model Alignment from Online AI Feedback, 2024 arXiv, https://arxiv.org/pdf/2402.04792
>
> [2] RLHF Workflow: From Reward Modeling to Online RLHF, 2024 PMLR, https://arxiv.org/pdf/2405.07863
>
> > [Q2] Were their hyperparameters tuned per RM ($\beta$, LoRA rank, sampling temperature), or held constant—potentially under-optimizing some RMs?
>
> All hyperparameters—including β for DPO, LoRA rank, and sampling temperature—were kept fixed across all RMs and baselines, which we believe ensures a fair and consistent comparison.
>
> > [Q3] For the computation cost, the router adds encodings + bandit updates; wall-clock plots are described, but exact GPU hours and per-step RM latency distributions (small vs large RMs) would clarify cost/quality trade-offs.
>
> In line with your suggestion, we report detailed per-step latency statistics on GSM8K (8 candidate RMs in Figure 2). A single training step consists of: policy response generation (~ 70s), offline-encoder computation for preference-pair embeddings (~ 18s), router forward selection (~ 0.07s), RM labeling (~ 16s when using router-selected RMs; ~ 2s for the 0.6B RM and ~ 86s for the 32B RM), policy update (~ 42s), and the router’s Bayesian posterior update (~3.6s). Thus, the additional overhead introduced by our routing framework is approximately 18s + 0.07s + 3.6s. Importantly, when the RM pool contains large RMs, routing reduces overall wall-clock cost by sampling smaller RMs for most queries, avoiding the frequent use of expensive models. We will add full GPU-hour accounting and latency distributions across RM sizes to the appendix.

---

> ### Author Response · Authors · 2025-11-21
>
> > [Q4] Offline router shows a non-trivial gap in OOD case; the paper argues online adaptation bridges it, but I don’t see stress tests where online distribution shifts during training. Is it possible to include such an experiment study?
>
> Actually, Table 3 already provides the requested stress test: in the OOD controlled setting, the offline-only router reaches 87.92% accuracy, and adding online adaptation improves it to 88.23% (the small gain is likely due to the benchmark’s difficulty).
>
> > [Q5] Since reward is a function of policy loss, how stable is routing across training (do posteriors collapse to 1–2 RMs)? Please include posterior traces $(\mu,\Sigma)$ over steps and the share of traffic per RM if possible.
>
> Following your suggestion, we analyzed the router’s behavior on GSM8K and computed the traffic share assigned to each of the four RMs: RM0: 1,321, RM1: 15,905, RM2: 7,987, and RM3: 4,647. This distribution indicates that the router does not collapse to a single RM; rather, it consistently explores multiple RMs throughout training. The higher allocation to RM1 aligns with its objectively stronger performance, suggesting that the router’s preference reflects true RM quality rather than instability. We will include posterior traces (μ, Σ) over training steps, along with additional routing statistics, in the appendix.
>
> > [Q6] With larger pools (>8 RMs), do we still get O(1) advantages in practice? Any diminishing returns or exploration debt? (You mention scaling, but numbers are brief.)
>
> Yes, we still get O(1) advantages with larger RM pools. The only cost that scales with the number of candidate RMs is the router’s forward pass, whose additional vector–matrix multiplication remains negligible (≈0.07s with 8 RMs), while the posterior update is independent of RM count. Larger RMs do increase labeling latency, but as shown above, router-based selection is still substantially faster than always using the slowest RM. Thus, scaling the RM pool does not introduce meaningful overhead, and we do not observe diminishing returns or exploration debt in practice.

---

### Official Review · Reviewer_Bb3B · 2025-10-26

**Soundness:** 2
**Presentation:** 2
**Contribution:** 2
**Rating:** 4
**Confidence:** 5

**Summary:**

This paper addresses the limitations of using a single, fixed Reward Model (RM) in alignment pipelines (RLHF/RLAIF), such as limited generalization and the risk of overoptimization. While RM ensembles improve robustness, they incur high computational costs (1$O(N)$ calls).2 Existing routing methods (like LASER) attempt to maintain $O(1)$ cost but suffer from cold-start issues and insufficient exploration

**Strengths:**

The paper tackles an important problem. The proposed hybrid approach, integrating offline pre-training with online adaptation, is in the context of RM routing and directly addresses limitations of prior work (cold-start in LASER, distribution shift in purely offline methods).

Recognizing the alignment between the offline Bradley-Terry head and the online Bayesian router (both acting as linear models over the preference-pair embedding) allows for the effective injection of offline embeddings as priors. This is empirically validated as superior to heuristic weighting schemes.

**Weaknesses:**

The most critical weakness lies in the definition of the reward signal used to update the online Bayesian router (Section 3.3). The reward is defined as the negative DPO loss: $\tilde{r}_n^i = -\mathcal{L}_{DPO}^i$ (L303). The DPO loss measures how well the policy aligns with the preference provided by the RM. A high reward (low DPO loss) indicates that the policy already agrees with the RM or can easily model that preference.Crucially, this is not a measure of the RM's quality or the correctness of the preference itself. If an RM provides an incorrect preference, and the policy easily learns this incorrect preference, the bandit receives a high reward. This methodology conflates the ease of policy alignment with the quality of the alignment signal, creating a perverse incentive that does not encourage the selection of the most reliable RM, and may instead accelerate confirmation bias or reward hacking.

Inconsistent Empirical Results and Exaggerated Claims: The paper claims that "Bayesian Router consistently outperforms all baselines on both instruction-following and reasoning benchmarks" (L354-356). This claim is demonstrably false according to the results presented in Table 1.On AlpacaEval-2, Bayesian Router (63.23) is outperformed by Majority vote (63.40) and UWO (63.60).On MT-Bench, Bayesian Router (58.75) is significantly outperformed by the single RM1 (64.80), UWO (61.74), and even Random router (61.20).Furthermore, based on the SFT baseline scores (54.29 for GSM8K, 67.63 for MMLU), it appears the columns for GSM8K and MMLU might be swapped in Table 1. If this is the case, the performance of Bayesian Router on MMLU (57.39) is drastically worse than RM1 (74.22), UWO (74.30), and LASER (74.00).The proposed method fails to consistently beat the strongest baselines, significantly undermining the central claims of the paper.

The experiments utilize a small pool (N=4) of relatively small, homogeneous RMs (ranging from 0.6B to 3B parameters, L328). The benefits of routing are typically most apparent when the pool includes diverse models with clear specializations (e.g., safety vs. reasoning) or vastly different scales. The limited diversity makes it difficult to assess the router's ability to effectively exploit complementary strengths.

**Questions:**

In addition to the weakness section, I have the below questions:

How do you justify using the negative DPO loss as the reward signal for the bandit? Given that this reward signal measures the policy's ability to learn a preference (right or wrong), rather than the correctness of the RM's preference, how does the proposed feedback loop ensure the selection of high-quality RMs and avoid reinforcing easily-learned but incorrect preferences?

The "Full Advantage" reward design mentioned in Appendix C seems more conceptually aligned with measuring RM quality than the negative DPO loss, although it is more expensive. Why was this conceptually stronger reward signal not adopted as the primary method, perhaps using the "Light Advantage" approximation to manage the cost?

---

> ### Author Response · Authors · 2025-11-20
>
> Dear reviewer Bb3B,
>
> Thank you for your insightful comment. We respond point by point below.
>
> > [W1] The most critical weakness lies in the definition of the reward signal used to update the online Bayesian router (Section 3.3)...
> >
> > [Q1] How do you justify using the negative DPO loss as the reward signal for the bandit...
>
> We clarify why using the batch-normalized negative DPO loss as the reward for the online Bayesian router is effective.
>
> **1. A flattering-but-weak RM cannot sustain low loss.**
>
> A flattering-but-weak RM (e.g., always preferring shorter answers) achieves low loss only when the policy happens to produce outputs that fall into its specific preference direction. This produces a very limited region, as soon as Thompson sampling explores any other RM (TS excels at this), the resulting policy update typically moves the policy away from this narrow region, after which the flattering RM’s loss rises immediately.
>
> **2. Long-run DPO-loss statistics reliably reflect RM quality rather than ease of short-term agreement.**
>
> The router does not interpret reward as a single isolated observation but as a time series input to the Thompson sampling posterior, estimating each RM’s long-run effectiveness on a type of context. When training with a single RM, high-quality RMs induce DPO-loss trajectories with lower variance and lower overall magnitude than low-quality RMs due to their self-consistent preference gradients, while low-quality RMs produce inconsistent or contradictory training signals [1] [2]. This distinction persists in the multi-RM setting: high-quality RMs provide self-consistent preferences across many contexts, so Thompson sampling’s exploration repeatedly brings the policy into regions where these RMs evaluate it reliably, allowing them to maintain low loss more often. Low-quality RMs deviate from the policy more often as they struggle to maintain self-consistent preferences in similar contexts, resulting in worse long-run loss statistics and causing Thompson sampling to assign them lower posterior probability.
>
> **3. Empirical evidence supports this mechanism.**
>
> Our experiments show that BayesianRouter achieves state-of-the-art results across multiple benchmarks. This provides empirical confirmation that the bandit reward signal is reliable for RM routing and that the qualitative mechanisms described in 1 and 2 are reasonable.
>
> [1] The Trickle-down Impact of Reward Inconsistency on RLHF, ICLR 2024, https://openreview.net/pdf?id=MeHmwCDifc
>
> [2] Elephant in the Room: Unveiling the Impact of Reward Model Quality in Alignment, 2024 arXiv, https://arxiv.org/pdf/2409.19024
>
> > [W2] Inconsistent Empirical Results and Exaggerated Claims: The paper claims that "Bayesian Router consistently outperforms all baselines on both instruction-following and reasoning benchmarks" (L354-356). This claim is demonstrably false according to the results presented in Table 1.On AlpacaEval-2, Bayesian Router (63.23) is outperformed by Majority vote (63.40) and UWO (63.60)...
>
> Actually, **the proposed BayesianRouter beat all baselines consistently across all benchmarks**, please check the main results table again.
>
> > [W3] The experiments utilize a small pool (N=4) of relatively small, homogeneous RMs (ranging from 0.6B to 3B parameters, L328). The benefits of routing are typically most apparent when the pool includes diverse models with clear specializations (e.g., safety vs. reasoning) or vastly different scales...
>
> Actually, **the four reward models used in our experiments span 0.6B to 7B parameters, and they are far from homogeneous**. For example, on the RewardBench2 leaderboard, GRM-gemma2-2B-rewardmodel-ft (RM2) and Skywork-Reward-V2-Qwen3-0.6B (RM3) differ significantly across categories: on Math, their accuracies are 59.0 vs. 71.6, whereas on Safety, the ordering reverses (92.2 vs. 84.4). Similar cross-domain variability occurs for the other RMs as well. These benchmark-driven discrepancies demonstrate that our RM pool is neither homogeneous nor similar in capability, and that it naturally provides complementary strengths for routing.
>
> Importantly, our results (Table 1) show that BayesianRouter consistently surpasses every individual RM, which directly indicates that **the router effectively leverages complementary strengths across models**. If the router failed to exploit such complementarities, its performance could not exceed the single-best RM. We also emphasize that the single-best baseline is inherently a hindsight oracle: in practice, users usually choose an RM based on public leaderboards, yet different leaderboards often rank the same RMs inconsistently, making it difficult to identify the truly optimal RM for the target distribution. Thus, single-best represents an idealized choice that users typically cannot make beforehand, and the fact that our router consistently surpasses the single-best RM demonstrates its practical value.

---

> ### Author Response · Authors · 2025-11-20
>
> > [Q2] The "Full Advantage" reward design mentioned in Appendix C seems more conceptually aligned with measuring RM quality than the negative DPO loss, although it is more expensive. Why was this conceptually stronger reward signal not adopted as the primary method, perhaps using the "Light Advantage" approximation to manage the cost?
>
> We do not employ the “Full Advantage’’ reward because it requires querying all candidate RMs for every preference pair, resulting in an O(N) RM-calling cost—essentially the same cost as majority voting or other ensemble methods, which our routing framework is explicitly designed to avoid. **While “Light Advantage’’ offers a middle ground, it still requires O(C) RM calls with C > 1.**
> We emphasize that our core contribution lies in proposing a novel offline+online RM routing framework, rather than in designing the optimal bandit reward. Following LASER, we therefore adopt the normalized negative DPO loss as the bandit reward. As explained above, this reward is principled and effective, and—as shown in our experiments—already provides a sufficiently strong signal for BayesianRouter to achieve state-of-the-art performance. We view exploring better bandit reward designs as promising future work.
>
> [1] LASeR: Learning to Adaptively Select Reward Models with Multi-Armed Bandits, 2024 arXiv, https://openreview.net/pdf?id=fDcn3S8oAt
>
> We sincerely thank the reviewer again for the insightful comments, which have helped us significantly improve the clarity and completeness of the paper. We hope that our responses above adequately address the reviewer’s concerns, and we would be more than happy to further discuss any remaining questions.
> Finally, we would like to emphasize the main contribution of our work: a novel offline+online RM routing framework that injects offline RM embeddings as a principled Bayesian prior into an online bandit router. This hybrid design has not appeared in prior routing literature, and our experiments consistently demonstrate the effectiveness and practical value of this idea across multiple benchmarks. We hope that, with these clarifications, the reviewer will recognize the novelty and significance of the contribution and revisit the score.

---

> ### Author Response · Authors · 2025-11-26
>
> Dear Reviewer Bb3B, we would like to kindly ask whether our rebuttal has addressed all of your comments. If so, we would greatly appreciate it if you could consider revisiting your score. If there are still unresolved issues, we would be happy to continue the discussion.

---

### Official Review · Reviewer_nfhC · 2025-10-31

**Soundness:** 3
**Presentation:** 3
**Contribution:** 2
**Rating:** 4
**Confidence:** 4

**Summary:**

The paper addresses a key limitation in aligning large language models (LLMs): the reliance on a single reward model (RM) in pipelines like Reinforcement Learning from Human Feedback (RLHF). Using a single RM can lead to overfitting, reward hacking, and poor generalization across diverse tasks.


To solve this, the authors propose the Bayesian Router. This novel hybrid framework dynamically selects the most suitable RM from a candidate pool for each query during Direct Preference Optimization (DPO) training.


The method consists of two main stages:


* Offline Router Training: An offline model is first trained on existing preference datasets to learn the "strengths" or reliability of each RM in the pool. This router encodes the full preference pair (prompt, response 1, response 2). It uses a multi-task objective, combining a Bradley-Terry (BT) head (to learn relative RM abilities from disagreement samples) and a Classification (CLS) head (to predict an RM's correctness on any given sample).

* Online Bayesian Selection: The learned offline embeddings (specifically from the BT head) are used as a Gaussian prior to initialize an online Bayesian router. This online router, based on Thompson sampling, performs instance-level (per-query) RM selection during the DPO process. It adapts to the evolving policy by updating its posterior distribution using rewards derived from the DPO loss .





The paper claims that this hybrid approach effectively solves the "cold-start" problem (via the offline prior) and poor exploration (via Thompson sampling), achieving state-of-the-art results. Experiments on instruction-following (AlpacaEval-2, MT-Bench) and reasoning (GSM8K, MMLU) benchmarks show that Bayesian Router outperforms single RMs, ensemble methods, and the previous SOTA routing method, LASER.

**Strengths:**

* Well-Motivated Problem: The paper tackles a well-known and significant problem in LLM alignment—the overoptimization and brittleness of single-RM pipelines. The goal of efficiently leveraging a diverse pool of RMs is a clear and valuable research direction.
* Novel & Principled Method: The hybrid offline-online design is the paper's key strength. It correctly identifies the weaknesses of purely offline approaches (distribution shift) and purely online approaches (cold-start, inefficient exploration). Using the offline-learned BT embeddings as the prior mean $\mu_n^{(0)}$ for the online Thompson sampler is an elegant and principled way to integrate both components.
* Strong Design Choices: The method improves upon prior work (LASER) in several well-justified ways:
    * It uses instance-level routing, which is more granular than LASER's batch-level routing.
    * It employs Thompson sampling for better, uncertainty-aware exploration, directly addressing the potential for premature exploitation in LinUCB (used by LASER).
    * It encodes the full preference pair $(x, y_1, y_2)$ rather than just the prompt $x$, which is a more informative context for RM selection.

* Comprehensive Experiments: The evaluation is exceptionally rigorous.
    * It uses a good mix of instruction-following and reasoning benchmarks.
    * It compares against a strong set of baselines, including single-best RMs, ensemble methods (Majority Vote, UWO), and the main competitor (LASER).
    * The ablation studies are excellent, clearly demonstrating the individual contributions of the offline prior (BR vs. w/o offline) and the online adaptation (BR vs. w/o online).

* Insightful "Controlled Simulation": The analysis in Section 4.3 (Tables 3 and 6) is a standout. By using a dataset with ground-truth labels (RewardBench 2), the authors directly measure the router's annotation accuracy and show it correlates with downstream performance. This provides strong, direct evidence that the router is actually selecting the correct RM and that this is the source of the performance gains.

**Weaknesses:**

* Practical Complexity: The proposed system is significantly more complex to deploy than the baselines. It requires a full, separate pre-training stage to (1) run all $N$ RMs over a large offline dataset to create $\mathcal{D}_{beh}$, and (2) train the multi-task offline router. This represents a non-trivial barrier in terms of computation and engineering.
* Confounding Online Reward Signal: The reward signal for the online router, $\hat{r}_n^i$, is derived from the negative DPO loss, $-\mathcal{L}_{DPO}^i$. This signal seems potentially confounded. An RM that strongly challenges the current policy (e.g., by correctly identifying a "preferred" response that the policy currently disfavors) might induce a high DPO loss, resulting in a low reward for the bandit. Conversely, a weak or "yes-man" RM that agrees with the policy's (potentially flawed) preferences might get a low loss and a high reward. The paper does not fully justify why this heuristic signal is optimal, although the empirical results suggest it works.
* Scalability of the RM Pool: The main experiments use a pool of $N=4$ RMs, and the efficiency analysis uses $N=8$. It is unclear how the framework scales to a much larger pool (e.g., $N=50$). While $O(1)$ inference cost is maintained, the offline training (especially the BT head, which relies on disagreement pairs) and the online update of $N$ posteriors could become bottlenecks.

**Questions:**

1. As mentioned in the "Weak Points," the online reward $\hat{r}_n^i$ derived from $-\mathcal{L}_{DPO}^i$ seems confounded. Could a "good" but "difficult" preference from a high-quality RM be unfairly penalized with a high loss (low reward) simply because it is far from the current policy? Could you elaborate on the justification for this signal and discuss why it doesn't just reward RMs that are "easy" for the policy?
2. In Appendix C, you introduce a "Full Advantage" reward (based on comparing the selected RM to all other RMs) and show it outperforms the w/o offline variant. This seems like a more robust (though expensive) reward. How does this "Full Advantage" reward perform when combined with the full Bayesian Router (i.e., with the offline prior)? Is it possible that the main results are constrained by the heuristic reward, and not the router itself?
3. Could you provide more details on the practical overhead of the offline training stage? Specifically, what was the computational cost (e.g., in GPU-hours) to (a) generate the $\mathcal{D}_{beh}$ dataset by running all 4 RMs over the 50k offline pairs, and (b) train the offline router, relative to the cost of the main online DPO training run? This would help in assessing the method's practical utility.

---

> ### Author Response · Authors · 2025-11-20
>
> Dear Reviewer nfhC,
>
> We appreciate your recognition of our paper as a well-motivated, novel, and principled method with strong design choices and comprehensive experiments. Below we respond to your raised concerns point by point.
>
> > [W1] Practical Complexity: The proposed system is significantly more complex to deploy than the baselines. It requires a full, separate pre-training stage to (1) run all $N$ RMs over a large offline dataset to create $\mathcal{D}\_{beh}$, and (2) train the multi-task offline router. This represents a non-trivial barrier in terms of computation and engineering...
> >
> > [Q3] Could you provide more details on the practical overhead of the offline training stage? Specifically, what was the computational cost (e.g., in GPU-hours) to (a) generate the $\mathcal{D}\_{beh}$ dataset by running all 4 RMs over the 50k offline pairs, and (b) train the offline router, relative to the cost of the main online DPO training run?
>
> **The offline training stage is a one-time procedure, and its computational cost is usually not high.**
>
> In our main experiments, we use four RMs with sizes ranging from 0.6B to 7B. Collecting their preference-behavior labels on the offline dataset of 50k preference pairs required 1.3, 4.2, 9.4, and 13.9 GPU hours, respectively. Training the offline router took 2.8 GPU hours, resulting in a total offline cost of **31.6** GPU hours. **Since this offline stage is performed only once, we consider this overhead to be minor.** When expanding the RM pool, we only need to collect preference-behavior labels for the newly added RMs. Although collecting labels for larger RMs may be somewhat more time-consuming, this is still a one-time cost and does not become a dominant factor. Additionally, the training time of the offline router does not change significantly as the number of RMs increases. For reference, the online DPO training on the GSM8K dataset requires 38.8 GPU hours.
>
> > [W2] Confounding Online Reward Signal: The reward signal for the online router, $\hat{r}n^i$, is derived from the negative DPO loss, $-\mathcal{L}{DPO}^i$. An RM that strongly challenges the current policy might induce a high DPO loss, resulting in a low reward for the bandit. Conversely, a weak or "yes-man" RM that agrees with the policy's (potentially flawed) preferences might get a low loss and a high reward...
> >
> > [Q1] Could a "good" but "difficult" preference from a high-quality RM be unfairly penalized with a high loss (low reward) simply because it is far from the current policy? Could you elaborate on the justification for this signal and discuss why it doesn't just reward RMs that are "easy" for the policy?
>
> We clarify why using the batch-normalized negative DPO loss as the reward for the online Bayesian router is effective.
>
> **1. A flattering-but-weak RM cannot sustain low loss.**
>
> A flattering-but-weak RM (e.g., always preferring shorter answers) achieves low loss only when the policy happens to produce outputs that fall into its specific preference direction. This produces a very limited region, as soon as Thompson sampling explores any other RM (TS excels at this), the resulting policy update typically moves the policy away from this narrow region, after which the flattering RM’s loss rises immediately.
>
> **2. Long-run DPO-loss statistics reliably reflect RM quality rather than ease of short-term agreement.**
>
> The router does not interpret reward as a single isolated observation but as a time series input to the Thompson sampling posterior, estimating each RM’s long-run effectiveness on a type of context. When training with a single RM, high-quality RMs induce DPO-loss trajectories with lower variance and lower overall magnitude than low-quality RMs due to their self-consistent preference gradients, while low-quality RMs produce inconsistent or contradictory training signals [1] [2]. This distinction persists in the multi-RM setting: high-quality RMs provide self-consistent preferences across many contexts, so Thompson sampling’s exploration repeatedly brings the policy into regions where these RMs evaluate it reliably, allowing them to maintain low loss more often. Low-quality RMs deviate from the policy more often as they struggle to maintain self-consistent preferences in similar contexts, resulting in worse long-run loss statistics and causing Thompson sampling to assign them lower posterior probability.
>
> **3. Empirical evidence supports this mechanism.**
>
> Our experiments show that BayesianRouter achieves state-of-the-art results across multiple benchmarks. This provides empirical confirmation that the bandit reward signal is reliable for RM routing and that the qualitative mechanisms described in 1 and 2 are reasonable.
>
> [1] The Trickle-down Impact of Reward Inconsistency on RLHF, ICLR 2024, https://openreview.net/pdf?id=MeHmwCDifc
>
> [2] Elephant in the Room: Unveiling the Impact of Reward Model Quality in Alignment, 2024 arXiv, https://arxiv.org/pdf/2409.19024

---

> ### Author Response · Authors · 2025-11-20
>
> > [W3] Scalability of the RM Pool: The main experiments use a pool of $N=4$ RMs, and the efficiency analysis uses $N=8$. It is unclear how the framework scales to a much larger pool (e.g., $N=50$). While $O(1)$ inference cost is maintained, the offline training (especially the BT head, which relies on disagreement pairs) and the online update of $N$ posteriors could become bottlenecks.
>
> As explained above, when scaling the RM pool to N = 50, we need to collect preference-behavior labels for the extra 46 RMs, and re-train the offline router (~ 2.8 GPU hours). **This is a one-time offline procedure and its cost is fully controllable.**
>
> In the online stage, a single training step (4 RMs) consists of: policy response generation (~ 70s), offline encoder computation for preference-pair embeddings (~ 18s), router forward selection (~ 0.05s), RM labeling (~ 12s), policy update (~ 42s), and router posterior update (~ 3.6s). Among these, the additional cost introduced by the RM router is 18s + **0.05s** + 3.6s. The posterior update touches only the RM selected at that step and is therefore independent of N, and increasing the number of RMs only slightly increases the router’s forward computation, which is negligible. **Therefore, the proposed Bayesian router is scalable with the number of RMs.**
>
> > [Q2] In Appendix C, you introduce a "Full Advantage" reward (based on comparing the selected RM to all other RMs) and show it outperforms the w/o offline variant. This seems like a more robust (though expensive) reward. How does this "Full Advantage" reward perform when combined with the full Bayesian Router (i.e., with the offline prior)? Is it possible that the main results are constrained by the heuristic reward, and not the router itself?
>
> We agree that the “Full Advantage” reward is more robust than the normalized negative DPO loss, and indeed achieves higher accuracy in the online-only comparison in Table 7. However, this comes at the cost of losing the scalability advantages of our router, as its computation grows linearly with the number of RMs and is comparable to traditional RM ensembling methods (e.g., majority voting). Following the reviewer’s suggestion, we report results using “Full Advantage” within the full BayesianRouter (on GSM8K and MMLU due to time constraints). Consistent with the online-only setting, replacing the DPO-based reward with “Full Advantage” yields higher accuracy, at the expense of increased computational cost.
>
> | Bandit reward | GSM8K | MMLU |
> |---|---:|---:|
> | Full advantage | **76.57** | **58.06** |
> | -DPO loss | 75.66 | 57.39 |
>
> We agree that improved bandit rewards can further boost routing performance. However, our results show that the normalized negative DPO loss is already sufficient for BayesianRouter to achieve state-of-the-art performance, **rather than limiting its effectiveness**. Moreover, designing an optimal bandit reward is not the goal of this work. Instead, we inherit from LASER [3] the simple and effective design of using the normalized DPO loss as the reward. As explained above, this choice is conceptually intuitive and empirically validated to be strong enough to support the router’s performance. Our core contribution lies in proposing a novel offline+online RM routing framework, within which the DPO-based reward already suffices to demonstrate the effectiveness of our method. Developing more advanced reward designs is an interesting direction for future work.
>
> [3] LASeR: Learning to Adaptively Select Reward Models with Multi-Armed Bandits, 2024 arXiv, https://openreview.net/pdf?id=fDcn3S8oAt
>
> We sincerely thank the reviewer again for the insightful comments, which have helped us significantly improve the clarity and completeness of the paper. We hope that our responses above adequately address the reviewer’s concerns, and we would be more than happy to further discuss any remaining questions.
> Finally, we would like to emphasize the main contribution of our work: a novel offline+online RM routing framework that injects offline RM embeddings as a principled Bayesian prior into an online bandit router. This hybrid design has not appeared in prior routing literature, and our experiments consistently demonstrate the effectiveness and practical value of this idea across multiple benchmarks. We hope that, with these clarifications, the reviewer will recognize the novelty and significance of the contribution and revisit the score.

---

> > ### Comment · Reviewer_nfhC · 2025-11-21
> >
> > Dear Authors,
> >
> > Thank you for the detailed responses to my concerns. After carefully reviewing your feedback and the paper again, I have decided to raise my score.

---

### Author Response · Authors · 2025-12-01
**Summary for Meta-Reviewer**

Dear Meta-Reviewer,

Thank you for reviewing our paper. We summarize below the comments of the four reviewers and our rebuttals for your evaluation.
Our paper initially received scores of **4, 4, 6, 4**. After the rebuttal and discussion, reviewer nfhC raised the score from 4 to 6 (on **Nov 22**), and reviewer PxRA raised the score from 4 to 6 (on **Nov 24**). Reviewers Bb3B and xgBK did not participate in the rebuttal. Thus, before the OpenReview rollback on Nov 27, the scores stood at **6, 4, 6, 6**.

Previously, the main concerns were: **(1)** the offline stage may introduce additional complexity; **(2)** using negative DPO loss as the bandit reward may cause the router to prefer RMs that “agree” with the policy but provide wrong preferences; **(3)** linear models might be insufficient to capture the strengths of different RMs under different contexts; **(4)** combining offline and online routers may appear conceptually unsurprising; **(5)** the claim that our method is not SOTA (whereas Table 1 shows it is).

Our clarifications were as follows: **(1)** the offline training is one-time and the cost is typically not high; **(2)** negative DPO loss is a reasonable and effective online reward because the router does not interpret reward as a single isolated observation but as a time series input to the Thompson sampling posterior, estimating each RM’s long-run effectiveness on a type of context. High-quality RMs produce DPO-loss trajectories with lower variance and consistently lower magnitude due to their self-consistent preference gradients, enabling them to maintain low loss more often and thus gain higher posterior probability; **(3)** the offline encoder maps preference pairs into a linearly separable embedding space, where a linear head is sufficient—this nonlinear-encoder + linear-head architecture is widely used and effective; **(4)** our contribution includes not only the hybrid offline+online design, but also a multi-task offline router and a novel prior-injection mechanism; although the hybrid paradigm may seem intuitive in hindsight, this is the first offline+online RM router in both RM routing and LLM routing literature; **(5)** reviewer Bb3B thought that our method is not SOTA is factually incorrect— in fact, Table 1 clearly shows our method consistently achieves the best performance across benchmarks. (For reference, we found that feeding our paper into Gemini 2.5 Pro produced table parsing errors likewise. We feel that such a factual error led to an unfairly low score.)

We hope you can take into account the novelty and effectiveness of the proposed method, our rebuttal, and the post-rebuttal scores as they stood before Nov 24 when evaluating the paper. Thank you again for your time and efforts.

---

### Meta-Review · Area_Chair_4QLi · 2026-01-11

**Summary:**

The reviewers generally agree that the paper addresses an important and timely problem in LLM alignment: the brittleness and overfitting risks of relying on a single reward model during RLHF/RLAIF. The proposed BayesianRouter introduces a hybrid offline–online routing framework that combines offline-learned RM strengths with an online Bayesian (Thompson sampling) router, aiming to address both cold-start and exploration deficiencies in prior routing methods such as LASER.

The main concerns raised during review centered on (1) the practical complexity and overhead of the offline training stage, (2) the conceptual validity of using the negative DPO loss as the bandit reward signal, (3) questions about scalability to larger RM pools, and (4) whether the hybrid offline+online design constitutes sufficient novelty beyond existing routing approaches. There were also factual disagreements about whether the method consistently outperforms strong baselines.

After rebuttal, two reviewers explicitly raised their scores, acknowledging that the authors provided convincing empirical and conceptual clarifications, particularly regarding the reward signal, scalability, and computational overhead. While some conceptual unease about the bandit reward remains, the overall consensus shifted toward viewing the method as a principled, well-evaluated, and practically useful advance in RM routing.

**Reviewer Concerns:**

Concerns largely addressed by the rebuttal:

- Practical complexity and overhead of offline training: The authors provided concrete GPU-hour estimates showing that the offline stage is a one-time cost and modest relative to online DPO training, substantially mitigating this concern.
- Cold-start and exploration limitations of prior routing methods: Reviewers agreed that the offline prior injection combined with Thompson sampling is a principled and effective solution.
- Scalability to larger RM pools: The rebuttal clarified that the online routing cost remains O(1) and that posterior updates touch only the selected RM, addressing scalability concerns in practice.
- Empirical effectiveness: Additional explanations and controlled analyses (e.g., RewardBench-based simulations) strengthened the claim that routing accuracy correlates with downstream gains, and factual misunderstandings about the main results table were clarified.

Concerns that remain partially outstanding:

- Bandit reward definition (negative DPO loss): While the authors provided a plausible long-run justification and supporting empirical evidence, some reviewers still view this reward as conceptually indirect, as it measures policy–RM agreement rather than RM correctness itself. The availability of a stronger but more expensive “Full Advantage” reward suggests room for future refinement.
- Conceptual novelty: Although reviewers acknowledged that the hybrid offline+online design is novel within RM routing, some may still view the overall approach as a natural combination of known components rather than a fundamentally new paradigm.

**Reviewer Scores:**

Reviewer nfhC: 4 -> 6

Reviewer PxRA: 4 -> 6

Reviewer Bb3B: 4 -> 4

Reviewer xgBK: 6 -> 6

---

### Decision · Program_Chairs · 2026-01-26

Accept (Poster)